# New Dammarane-Type Triterpenoid Saponins from *Panax notoginseng* Leaves and Their Nitric Oxide Inhibitory Activities

**DOI:** 10.3390/molecules25010139

**Published:** 2019-12-29

**Authors:** Fan Sun, Jingya Ruan, Wei Zhao, Ying Zhang, Guilin Xiang, Jiejing Yan, Mimi Hao, Lijie Wu, Yi Zhang, Tao Wang

**Affiliations:** 1Tianjin Key Laboratory of TCM Chemistry and Analysis, Tianjin University of Traditional Chinese Medicine, 10 Poyanghu Road, West Area, Tuanbo New Town, Jinghai District, Tianjin 301617, China; sf18435165322@163.com (F.S.); Ruanjy19930919@163.com (J.R.); zhaowei126123@126.com (W.Z.); wulj0816@163.com (L.W.); 2Institute of TCM, Tianjin University of Traditional Chinese Medicine, 10 Poyanghu Road, West Area, Tuanbo New Town, Jinghai District, Tianjin 301617, China; zyingtzy@163.com (Y.Z.); 17320072093@163.com (J.Y.); haomimi126@126.com (M.H.); 3WenshanMiaoxiangSanqi Limited Company, South KaihuaRoad, Wenshan 663000, China; xiangguilin@126.com

**Keywords:** *Panax notoginseng* leaves, dammarane-type triterpenoid saponins, notoginsenosides NL, RAW 246.7 cell, anti-inflammatory activity

## Abstract

Inflammation is a very common and important pathological process that can cause many diseases. The discovery of anti-inflammatory drugs and the treatment of inflammation are particularly essential. Dammarane-type triterpenoid saponins (PNS) were demonstrated to show anti-inflammatory effects in the leaves of *Panax notoginseng*. Chromatographies and spectral analysis methods were combined to isolate and identify PNS. Moreover, the nitric oxide (NO) inhibitory activities of all compounds were examined in lipopolysaccharide (LPS)-stimulated RAW264.7 cells. As a result, eleven new dammarane-type triterpenoid saponins, notoginsenosides NL-A_1_–NL-A_4_ (**1**–**4**), NL-B_1_–NL-B_3_ (**5**–**7**), NL-C_1_–NL-C_3_ (**8**–**10**), and NL-D (**11**) were isolated, and their structures were identified by using various spectrometric techniques and chemical reactions. Among them, compounds **4** and **11** were characterized by the malonyl substitution at 3-position. The 3-malonyl substituted dammarane-type terpennoids were first obtained from natural products. In addition, compounds **1**, **2**, **5**, **6**, and **8**–**10** were found to play an important role in suppressing NO levels at 50 μM, without cytotoxicity. All inhibitory activities were found to be dose-dependent.

## 1. Introduction

*Panax notoginseng* (Burk.) F. H. Chen is generally known as San qi in Chinese. As recorded in Chinese medical book “Compendium of Material Medica”, its root was traditionally used in Asia for the treatment of trauma, body pain, inflammation, and cardiovascular diseases since ancient times [1]. Due to the excellent medical property of *P. notoginseng*, medicines that contain this leaf are commercially available and are widely applied in clinics in China. For example, *P. notoginseng* was found to be the main ingredient in Xuesaitong injections and Xuesaitong capsules [2], Yun-Nan-Bai-Yao [3], as well as Xueshuantong injections and Xueshuantong capsules. As is known, the triterpenoid saponins in it largely contributes to the above-mentioned biological activities [2,3].

Since the ability of *P. notoginseng* to adapt to the environment declined gradually through hundreds of years of cultivation, the problem of continuous cropping became more and more prominent, which resulted in the decrease of *P. notoginseng* root yields [4]. On the other hand, the harvest of *P. notoginseng* root required long growth periods. Research has showed that *P. notoginseng* leaves are rich in dammarane-type triterpenoid saponins (PNS) [5,6,7,8,9,10], suggesting that the leaves could be a possible replacement of the roots. In order to expand the utilization of *P. notoginseng* resources, the development and application of its leaves have gradually attracted the attention and interest of scholars.

Inflammation is a very common and important pathological process that can cause many diseases [11]. The discovery of anti-inflammatory drugs and the treatment of inflammation are particularly essential. PNS were demonstrated to show anti-inflammatory effects in *P. notoginseng* [3].

Herein, chromatographies and spectral analysis methods were combined to isolate and identify PNS from *P. notoginseng* leaves. Moreover, the inhibitory activities of obtained PNS against nitric oxide (NO) production in RAW 264.7 cells induced by lipopolysaccharide (LPS) were measured.

## 2. Results and Discussion

The 50% EtOH extract of *P. notoginseng* leaves was isolated by D101 macroporous resin column chromatography (CC), and was eluted with H_2_O and 95% EtOH, successively. The obtained 95% EtOH eluate was separated by CCs such as silica gel, Sephadex LH-20, and preparative high-performance liquid chromatography (pHPLC), and eleven new dammarane-type triterpenoid saponins, notoginsenosides NL-A_1_–NL-A_4_ (**1**–**4**), NL-B_1_–NL-B_3_ (**5**–**7**), NL-C_1_–NL-C_3_ (**8**–**10**), and NL-D(**11**) (Figure 1) were yielded.

Notoginsenoside NL-A_1_ (**1**) was isolated as a white powder with a negative optical rotation (${\left[\alpha \right]}_{D}^{25}$ −1.8, MeOH). Its molecular formula, C_47_H_80_O_19_ (*m*/*z* 947.52405 [M − H]^−^; calcd. for C_47_H_79_O_19_, 947.52101) was measured on negative-ion ESI-Q-Orbitrap MS. The IR spectrum showed the absorption bands assignable to hydroxyl (3395 cm^−1^), olefin (1645 cm^−1^), and ether (1078 cm^−1^) functions, respectively. Acid hydrolysis of **1** followed by HPLC analysis confirmed the presence of d-glucose and l-arabinose [12]. The ^1^H and ^13^C-NMR (Table 1) spectra of **1** displayed the signals of two β-d-glucopyranosyls [δ 4.95 (1H, d, *J* = 8.0 Hz, H-1′), 5.18 (1H, d, *J* = 8.0 Hz, H-1″)], and one α-l-arabinofuranosyl [δ 5.66 (1H, d, *J* = 1.5 Hz, H-1‴)]. Its ^13^C-NMR spectrum showed forty-seven signals. After subtracting the seventeen carbon resonances that belonged to the sugar units, the remaining thirty resonances were attributable to a triterpene skeleton. In the ^1^H-NMR spectrum, eight signals could be assigned to methyls [δ 0.81, 0.90, 1.00, 1.02, 1.32 (3H each, all s, H_3_-19, 30, 29, 18, and 28), and 1.61 (9H, s, H_3_-21, 26, and 27)], two signals belonged to oxygenated methylene [δ 3.36 (1H, dd, *J* = 4.0, 11.5 Hz, H-3), 4.02 (1H, m, H-12)], and the signals for one *trans*-olefin [δ 6.11 (1H, d, *J* = 16.0 Hz, H-24), 6.16 (1H, ddd, *J* = 5.5, 8.0, 16.0 Hz, H-23)] indicated that **1** was a dammarane-type triterpene saponin derivative. In order to solve the problem of overlapping for the three glycosyl groups, HSQC-TOCSY experiment was performed. In the HSQC-TOCSY spectrum, correlations were found between the following proton and carbon pairs: δ_H_ 4.95 (H-1′) and δ_C_ 71.8 (C-4′), 75.7 (C-2′), 78.7 (C-3′), 107.1 (C-1′); δ_H_ 4.42, 4.62 (H_2_-6′) and δ_C_ 63.0 (C-6′), 71.8 (C-4′), 78.4 (C-5′); δ_H_ 5.18 (H-1″) and δ_C_ 71.9 (C-4″), 75.1 (C-2″), 78.8 (C-3″), 98.2 (C-1″); δ_H_ 4.13, 4.66 (H_2_-6″) and δ_C_ 68.3 (C-6″), 71.9 (C-4′), 76.4 (C-5″); δ_H_ 5.66 (H-1‴) and δ_C_ 83.3 (C-2‴), 110.0 (C-1‴); δ_H_ 4.87 (H-2‴) and δ_C_ 78.9 (C-3‴), 83.3 (C-2‴), 85.9 (C-4‴), 110.0 (C-1‴); δ_H_ 4.21, 4.31 (H_2_-6‴) and δ_C_ 62.7 (C-5‴), 78.9 (C-3‴), 83.3 (C-2‴), 85.9 (C-4‴). In conjunction with the HSQC spectrum, the spectroscopic data of the above-mentioned three glycosyls were assigned. According to the proton and proton correlations observed in its ^1^H–^1^H COSY spectrum (Figure 2), seven moieties written in bold lines were denoted. Moreover, its planar structure of was clarified by the correlations from H_3_-18 to C-7–9, C-14; H_3_-19 to C-1, C-5, C-9, C-10; H_3_-21 to C-17, C-20, C-22; H_3_-26 and C-24, C-25, C-27; H_3_-27 to C-24–26; H_3_-28 to C-3−5, C-29; H_3_-29 to C-3−5, C-28; H_3_-30 to C-8, C-13–15; δ_H_ 4.95 (H-1′) to δ_C_ 88.8 (C-3); δ_H_ 5.18 (H-1″) to δ_C_ 83.2 (C-20); δ_H_ 5.66 (H-1‴) to δ_C_ 68.3 (C-6″) displayed in its HMBC spectrum. The resonance of C-25 were shifted downfield by about 11 ppm, as compared with notoginsenoside Fh5 with 3β,12β,20(*S*),25-tetrahydroxydammar-23-ene as aglycone [6], indicating that C-25 of compound **1** was substituted by the hydroperoxyl group. Moreover, since the chemical shifts of notoginsenoside NL-A_1_ (**1**) aglycone were identical to those of the known compound, 3β,12β,20*S*-trihydroxy-25-hydroperoxydammar-23-ene-3-*O*-[β-d-glucopyranosyl(1→2)-β-d-glucopyranosyl]-20-*O*-[β-d-xylopyranosyl(1→6)]-β-d-glucopyranoside [13], its aglycone was determined to be 3β,12β,20*S*-trihydroxy-25-hydroperoxydammar-23-ene.

Both notoginsenosides NL-A_2_ (**2**) and NL-A_3_ (**3**) were obtained as a white powder. The ^1^H and ^13^C-NMR (Table 2 and Table 3) and 2D-NMR spectra, including ^1^H–^1^H COSY, HSQC, HSQC-TOCSY, and HMBC suggested **2** and **3** had the same aglycone, 3β,12β,20*S*-trihydroxy-25-hydroperoxydammar-23-ene as that of **1**. Meanwhile, compound **2** possessed the same molecular formula, C_47_H_80_O_19_ (*m*/*z* 947.51996 [M − H]^−^; calcd. for C_47_H_79_O_19_, 947.52101) as **1**. The main difference between **2** and **1** was that the α-l-arabinofuranosyl group in **1** was replaced for β-d-xylopyranosyl group [δ 5.00 (1H, d, *J* = 7.5 Hz, H-1‴)] in **2**. As compared to compound **1**, one more β-d-glucopyranosyl [δ 5.50 (1H, d, *J* = 7.5 Hz, H-1″)] and one more β-d-xylopyranosyl [δ 5.39 (1H, d, *J* = 7.0 Hz, H-1‴)] presented in **3**, which was clarified by its ESI-Q-Orbitrap MS determination (*m*/*z* 1241.61584 [M − H]^−^; calcd. for C_58_H_97_O_28_, 1241.61609). Meanwhile, as compared to **1**, δ_C_-1′ and δ_C_-2′ of **3** were shifted upfield and downfield by [δ_C_ 75.7 (C-2′), 107.1 (C-1′) for **1**; δ_C_ 82.9 (C-2′), 104.8 (C-1′) for **3**], respectively, which suggested C-2′ in **3** might be substituted by the glycosyl group. The cross peaks from δ_H_ 4.98 (H-1′) to δ_C_ 88.8 (C-3); δ_H_ 5.20 (H-1″) to δ_C_ 83.2 (C-20); δ_H_ 5.00 (H-1‴) to δ_C_ 69.9 (C-6″) were observed in **2**; while the long range correlations from δ_H_ 4.92 (H-1′) to δ_C_ 89.0 (C-3); δ_H_ 5.50 (H-1″) to δ_C_ 82.9 (C-2′); δ_H_ 5.39 (H-1‴) to δ_C_ 84.6 (C-2″); δ_H_ 5.17 (H-1⁗) to δ_C_ 83.2 (C-20); δ_H_ 5.64 (H-1′′′′′) to δ_C_ 68.4 (C-6⁗) were found (Figure 2) in **3**. The structures of **2** and **3** were thus elucidated. Using methods similar as **1**, the NMR data of each sugar moieties in **2** and **3** were assigned in detail.

Notoginsenoside NL-A_4_ (**4**) was a white powder. Its molecular formula was determined to be C_44_H_72_O_17_, on the basis of the ESI-Q-Orbitrap MS (*m*/*z* 871.46875 [M − H]^−^; calcd. for C_44_H_71_O_17_, 871.46858) and NMR data (Table 4) analysis. The IR spectrum showed absorption bands at 3379, 1718, 1646, and 1080 cm^−1^, corresponding to hydroxyl, carbonyl, olefin, and ether groups, respectively. By comparison, the ^1^H and ^13^C-NMR spectra with those of **1**, **2**, and **3**, the aglycone of **4** was also proposed to be 3β,12β,20*S*-trihydroxy-25-hydroperoxydammar-23-ene. Its ^1^H and ^13^C-NMR spectra displayed the signals that could be assigned to one β-d-glucopyranosyl [δ 5.20 (1H, d, *J* = 7.0 Hz, H-1″)] and one α-l-arabinopyranosyl [δ 5.02 (1H, d, *J* = 6.0 Hz, H-1‴)]. Forty-four signals were observed in its ^13^C-NMR spectrum. Except for the signals that belonged to the above-mentioned aglycone and glycosyl groups, the other three signals could be assigned to two ester carbonyls [δ_C_ 168.3 (C-1′), 169.9 (C-3′)] together with one methylene [δ_C_ 43.8 (C-2′)]. Combining with the chemical shift [δ 3.83 (2H, s, H_2_-2′)] of methylene proton, the presence of malonyl was deduced, which was confirmed by the cross peaks from δ_H_ 3.83 (H_2_-2′) to δ_C_ 168.3 (C-1′), 169.9 (C-3′). Moreover, the covalent connectivities of above-mentioned moieties were established by the long-range correlations from δ_H_ 4.81 (1H, dd, *J* = 4.5, 11.5 Hz, H-3) to δ_C_ 168.3 (C-1′); δ_H_ 5.20 (H-1″) to δ_C_ 83.3 (C-20); δ_H_ 5.02 (H-1‴) to δ_C_ 69.0 (C-6″) (Figure 2). The NMR data for aglycone of **4** were remarkably reminiscent of those of compounds **1**, **2**, and **3**, besides C-2−4. The 3-OH group was established as β-oriented (equatorial) from the NOESY correlations beween δ_H_ 4.01 (H-12) and δ_H_ 1.38 (H-9); δ_H_ 0.78 (H-5) and δ_H_ 1.38 (H-9), 4.81 (H-3) and vicinal coupling constants of H-3 with H_2_-2 (^3^*J* = 4.5, 11.5 Hz). Then, its aglycone was also confirmed to be 3β,12β,20*S*-trihydroxy-25-hydroperoxydammar-23-ene. On the basis of the above-mentioned evidence, the structure of notoginsenoside NL-A_4_ (**4**) was determined.

Notoginsenoside NL-A_4_ (**4**) was a white powder. Its molecular formula was determined to be C_44_H_72_O_17_, on the basis of the ESI-Q-Orbitrap MS (*m*/*z* 871.46875 [M − H]^−^; calcd. for C_44_H_71_O_17_, 871.46858) and NMR data (Table 4) analysis. The IR spectrum showed absorption bands at 3379, 1718, 1646, and 1080 cm^−1^, corresponding to hydroxyl, carbonyl, olefin, and ether groups, respectively. By comparison, the ^1^H and ^13^C-NMR spectra with those of **1**, **2**, and **3**, the aglycone of **4** was also proposed to be 3β,12β,20*S*-trihydroxy-25-hydroperoxydammar-23-ene. Its ^1^H and ^13^C-NMR spectra displayed the signals that could be assigned to one β-d-glucopyranosyl [δ 5.20 (1H, d, *J* = 7.0 Hz, H-1″)] and one α-l-arabinopyranosyl [δ 5.02 (1H, d, *J* = 6.0 Hz, H-1‴)]. Forty-four signals were observed in its ^13^C-NMR spectrum. Except for the signals that belonged to the above-mentioned aglycone and glycosyl groups, the other three signals could be assigned to two ester carbonyls [δ_C_ 168.3 (C-1′), 169.9 (C-3′)] together with one methylene [δ_C_ 43.8 (C-2′)]. Combining with the chemical shift [δ 3.83 (2H, s, H_2_-2′)] of methylene proton, the presence of malonyl was deduced, which was confirmed by the cross peaks from δ_H_ 3.83 (H_2_-2′) to δ_C_ 168.3 (C-1′), 169.9 (C-3′). Moreover, the covalent connectivities of above-mentioned moieties were established by the long-range correlations from δ_H_ 4.81 (1H, dd, *J* = 4.5, 11.5 Hz, H-3) to δ_C_ 168.3 (C-1′); δ_H_ 5.20 (H-1″) to δ_C_ 83.3 (C-20); δ_H_ 5.02 (H-1‴) to δ_C_ 69.0 (C-6″) (Figure 2). The NMR data for aglycone of **4** were remarkably reminiscent of those of compounds **1**, **2**, and **3**, besides C-2−4. The 3-OH group was established as β-oriented (equatorial) from the NOESY correlations beween δ_H_ 4.01 (H-12) and δ_H_ 1.38 (H-9); δ_H_ 0.78 (H-5) and δ_H_ 1.38 (H-9), 4.81 (H-3) and vicinal coupling constants of H-3 with H_2_-2 (^3^*J* = 4.5, 11.5 Hz). Then, its aglycone was also confirmed to be 3β,12β,20*S*-trihydroxy-25-hydroperoxydammar-23-ene. On the basis of the above-mentioned evidence, the structure of notoginsenoside NL-A_4_ (**4**) was determined.

Notoginsenoside NL-B_1_ (**5**) was obtained as a white powder with positive optical rotation (${\left[\alpha \right]}_{D}^{25}$ +2.2, MeOH). Its IR spectrum exhibited absorption maxima at 3379, 1649 and 1076 cm^−1^, accounting for the hydroxyl, olefinic bond, and ether function, respectively. d-glucose and l-arabinose were detected from its acid hydrolysis product [12]. Its ^1^H and ^13^C-NMR (Table 5) spectra revealed the difference only in H-23–27 [δ 1.56, 1.57 (3H each, both s, H_3_-26, and 27), 6.09 (1H, d, *J* = 15.5 Hz, H-24), 6.23 (1H, ddd, *J* = 5.5, 8.5, 15.5 Hz, H-23)] and C-23–27 [δ_C_ 30.6 (C-27), 30.7 (C-26), 70.0 (C-25), 122.7 (C-23), 142.2 (C-24)], as compared to that of compound **1**. The ESI-Q-Orbitrap MS analysis result indicated its molecular formula was C_47_H_80_O_18_ (*m*/*z* 931.52838 [M − H]^−^; calcd. for C_47_H_79_O_18_, 931.52609), an oxygen atom disappeared when comparing it with **1**, suggesting that C-25 of **5** was substituted by the hydroxyl group. The cross peaks from δ_H_ 1.57 (H_3_-26) to δ_C_ 30.6 (C-27), 70.0 (C-25), 142.2 (C-24); δ_H_ 1.56 (H_3_-27) to δ_C_ 30.7 (C-26), 70.0 (C-25), 142.2 (C-24), clarified the above-mentioned description. The NMR data of its aglycone were likable to those of the known compound, 3-*O*-β-d-xylopyranosyl-(1→2)-β-d-glucopyranosyl-(1→2)-β-d-glucopyranosyl-3β,12β,20(*S*),25-tetrahydroxydammar-23-ene-20-*O*-β-d-glucopyranoside-(1→6)-β-d-glucopyranoside [6]. Thus, its aglycone was elucidated to be 3β,12β,20(*S*),25-tetrahydroxydammar-23-ene. Finally, the linkage positions between sugar and sugar, as well as sugar and aglycone were clarified by the long-range correlation observations from δ_H_ 4.96 (H-1′) to δ_C_ 89.0 (C-3); δ_H_ 5.18 (H-1″) to δ_C_ 83.3 (C-20); and δ_H_ 5.65 (H-1‴) to δ_C_ 68.4 (C-6″).

The molecular formula of both notoginsenosides NL-B_2_ (**6**) and NL-B_3_ (**7**) were determined to be C_58_H_98_O_27_, on the basis of the MS analysis (**6**: *m*/*z* 1225.62134 [M − H]^−^, **7**: *m*/*z* 1225. 61841 [M − H]^−^, both calcd. for C_58_H_97_O_27_, 1225.62117). Based on their ^1^H- and ^13^C-NMR (Table 6 and Table 7), ^1^H–^1^H COSY, HSQC, HMBC, and HSQC-TOCSY spectral characteristics, the aglycones of both **6** and **7** were identified as 3β,12β,20(*S*),25-tetrahydroxydammar-23-ene, too. Moreover, by comparing to **3**, we could deduce that **6** had the same glycosyl moieties as **3**. d-glucose, l-arabinose, and d-xylose were yielded from its acid hydrolysis reaction [12]. Moreover, the long-range correlations from δ_H_4.92 (H-1′) to δ_C_ 89.0 (C-3); δ_H_5.50 (H-1″) to δ_C_ 83.0 (C-2′); δ_H_5.39 (H-1‴) to δ_C_ 84.6 (C-2″); δ_H_5.16 (H-1⁗) to δ_C_ 83.4 (C-20); and δ_H_5.64 (H-1′′′′′) to δ_C_ 68.4 (C-6⁗) were observed in the HMBC experiment (Figure 2). Those results confirmed that the glycosyl moieties of **6** were matched with **3**. Meanwhile, only d-glucose and d-xylose were detected from the acid hydrolysis product of **7**. In addition, the cross peaks displayed in its HMBC spectrum (Figure 2) denoted that the C-20-substituted glycosyl in **7** was β-d-xylopyranosyl(1→6)-β-d-glucopyranosyl [δ 4.98 (1H, d, *J* = 7.2 Hz, H-1′′′′′), 5.18 (1H, d, *J* = 7.8 Hz, H-1⁗)].

Notoginsenoside NL-C_1_ (**8**), white powder, exhibited a molecular ion peak [M − H]^−^ at *m*/*z* 929.51184 (calcd. for C_47_H_77_O_18_, 929.51044) in ESI-Q-Orbitrap MS spectrum, which matched to the molecular formula, C_47_H_78_O_18_, confirmed by the NMR spectral data (Table 8). Its IR absorption bands indicated the presence of hydroxyl (3387 cm^−1^), α,β-unsaturated ketone carbonyl (1669 cm^−1^), olefin (1637 cm^−1^), and an ether function (1077 cm^−1^). Its ^1^H- and ^13^C-NMR spectra showed that the signals belonged to two β-d-glucopyranosyl [δ4.95 (1H, d, *J* = 7.5 Hz, H-1′), 5.10 (1H, d, *J* = 8.0 Hz, H-1″)], and one α-l-arabinofuranosyl [δ5.71 (1H, d, *J* = 1.5 Hz, H-1‴)]. Its ^13^C-NMR spectrum displayed forty-seven carbons including thirty for aglycone and seventeen for sugar moieties. The ^1^H and ^13^C-NMR spectra denoted that **8** was a dammarane-type triterpene saponin derivative. Compared to **1**, the chemical shifts of proton and carbon for the side chain changed significantly, as follows—one disubstituted terminal olefin signals [δ5.78, 6.37 (1H each, both br. s, H_2_-26)] appeared, while the signals of one methyl and one *trans*-olefinic bond disappeared in its ^1^H-NMR spectrum; meanwhile, an additional terminal olefin [δ_C_ 125.5 (C-26), 144.4 (C-25)] and additional carbonyl group [δ_C_202.6 (C-24)] signals were displayed in its ^13^C-NMR spectrum, but one methyl, one *trans*-olefinic bond, as well as oxygenated quaternary carbon signals disappeared. The ^1^H–^1^H COSY experiment on **8** indicated the presence of partial structures written in bold lines, as shown in Figure 2. Finally, in the HMBC experiment, the following long-range correlations from proton to carbon pairs were found: δ_H_ 5.78, 6.37 (H_2_-26) to δ_C_ 17.8 (C-27), 144.4 (C-25), 202.6 (C-24); δ_H_ 1.85 (H_3_-27) to δ_C_ 125.5 (C-26), 144.4 (C-25), 202.6 (C-24), then the conjunction position of carbonyl and terminal olefin were clarified. Since the chemical shifts of its aglycone were identical to those of ginsenoside III [14], its aglycone of was identified as 3β,12β,20(*S*)-trihydroxy-dammar-25-en-24-one. Consequently, the structure of notoginsenoside NL-C_1_ (**8**) was elucidated.

Notoginsenoside NL-C_2_ (**9**) was obtained as a white powder with positive optical rotation (${\left[\alpha \right]}_{D}^{25}$ +1.8, MeOH). The same molecular formula, C_47_H_78_O_18_ (m/z 929.51221 [M − H]^−^; calcd. for C_47_H_77_O_18_, 929.51044) as that of **8** was revealed by the 0ESI-Q-Orbitrap MS analysis. After acid hydrolysis, d-glucose and d-xylose were given [12]. Its ^1^H and ^13^C-NMR (Table 9) spectra were similar to those of **8**. The main differences between them were one β-d-xylopyranosyl [δ4.91 (1H, d, *J* = 7.5 Hz, H-1‴)] appeared, and one α-l-arabinofuranosyl disappeared in **9**. Finally, the β-d-xylopyranosyl was clarified to have connected with 6-position of β-d-glucopyranosyl substituted at C-20 by cross peak from δ_H_ 4.91 (H-1‴) to δ_C_ 70.1 (C-6″), showed in HMBC (Figure 2).

The molecular formula of notoginsenoside NL-C_3_(**10**) was measured to be C_53_H_88_O_23_ (*m*/*z* 1091.56445 [M − H]^−^; calcd. for C_53_H_87_O_23_, 1091.56327) on an ESI-Q-Orbitrap MS spectrometer. Compared with **9**, one more group data of β-d-glucopyranosyl [δ 5.39 (1H, d, *J* = 7.5 Hz, H-1″)] was presented in **10** (Table 10). Moreover, the carbon signals of C-1′ and C-2′ significantly shifted to higher and lower field [δ_C_ 75.8 (C-2′), 107.0 (C-1′) for **9**; δ_C_ 83.5 (C-2′), 105.1 (C-1′) for **10**], respectively, which suggested that C-2′ was displaced by the glycosyl group. The long-range correlation from δ_H_ 5.39 (H-1″) to δ_C_ 83.5 (C-2′) displayed in its HMBC experiment confirmed the above-mentioned β-d-glucopyranosyl linked with C-2′.

Notoginsenoside NL-D (**11**) was a white powder with positive optical rotation (${\left[\alpha \right]}_{D}^{25}$ +29.6, MeOH). The molecular formula, C_44_H_72_O_16_ of **11** was determined by ESI-Q-Orbitrap MS (*m*/*z* 855.47388 [M − H]^−^; calcd. for C_44_H_71_O_16_, 855.47366). The IR absorptions at 3382, 1717, 1649, and 1075 cm^−1^ indicated the presence of OH, CO, olefin, and an ether group. The ^1^H and ^13^C-NMR (Table 11) and 2D NMR spectra, including ^1^H–^1^H COSY, HSQC, and HMBC suggested that **11** had the same sugar moiety, β-d-arabinopyranosyl(1→6)-β-d-glucopyranosyl [δ 5.00 (1H, d, *J* = 5.5 Hz, H-1‴), 5.12 (1H, d, *J* = 7.5 Hz, H-1″)] and malonyl [δ_C_ 43.8 (C-2′), 168.4 (C-1′), 170.6 (C-3′)] as **4**. Forty-four signals were observed in its ^13^C-NMR spectrum. Except for the signals belonging to the above-mentioned moieties, the other 30 indicated that **11** was a triterpene saponin. Its ^1^H-NMR spectrum showed eight methyl signals at δ0.93, 0.97, 1.05, 1.35, 1.62, 1.63, 1.66, and 1.76 (3H each, all s, H_3_-19, 30, 18, 29, 26, 21, 27, 28), three methines bearing an oxygen function at δ4.19 (1H, m, H-12), 4.32 (1H, m, overlapped, H-6), and 4.88 (1H, dd, *J* = 4.0, 10.5 Hz, H-3), one trisubstituted olefin signal at δ 5.32 (1H, t, *J* = 6.5 Hz, H-24). In its HMBC experiment, long-range correlations were observed from the following proton to carbon pairs: H-3 to C-1′; H_3_-18 to C-7−9, C-14; H_3_-19 to C-1, C-5, C-9, C-10;H_3_-21 to C-17, C-20, C-22; H_3_-26 to C-24, C-25, C-27; H_3_-27 to C-24−26;H_3_-28 to C-3−5, C-29; H_3_-29 to C-3−5, C-28; and H_3_-30 to C-8, C-13−15; H-1″ to C-20. Then, its planar structure was elucidated, which was a protopanaxatriol type saponin. The chemical shifts for proton and carbon of the B–D ring and the side chain were matched closely with those of ginsenoside U [15]. Since they were more different in 1–4 positions of two compounds, they might have been affected by the configuration of C-3 or a malonyl substitution. Finally, the NOE correlations between δ_H_ 4.19 (H-12) and δ_H_ 1.49 (H-9); δ_H_ 1.18 (H-5) and δ_H_ 1.49 (H-9), 4.88 (H-3) displayed in its NOESY experiment (Figure 2) denoted that H-3 showed α-orientation. Thus, the structure of notoginsenoside NL-D (**11**) was determined.

Moreover, in order to clarify the anti-inflammatory effects of **1**–**11**, their inhibitory activities on LPS-stimulated NO release were measured using the method reported previously in [16].

Before the experiment, dimethyl thiazolyl diphenyl tetrazolium (MTT) assay was used to test the cytotoxicities of **1**–**11**. It was found that compounds **1**, **2**, **5**, **6**, **8**–**10** displayed no significant cytotoxicity at 50 μM concentration (Appendix A). Then, under this concentration, in vitro potential anti-inflammatory effects of all isolates were investigated. As a result, all of them exhibited significant inhibitory effects of the NO release (Table 12).

Anti-inflammatory activities of **1**, **2**, **5**, **6**, and **8**–**10** suggested that the number and type of substituted glycosyls mattered a lot for the anti-inflammatory activity of triterpenoids—the activities of β-d-xylopyranosyl-substituted triterpenoids were higher than the ones tha were α-l-arabinofuranosyl-substituted (**1** vs. **2**; **8** vs. **9**); Meanwhile, with an increasing number of β-d-glucopyranosyl, a stronger activity appeared (**9** vs. **10**). Additionally, the different side chains of C-20 displayed a strong effect on their activities (**1** vs. **8**; **2** vs. **9**).

Moreover, a dose-dependent experiment was performed for the above-mentioned compounds at concentrations of 10, 25, and 50 μM, respectively. All of them were found to inhibit NO release from RAW264.7 cells in a dose-dependent manner (Figure 3).

## 3. Materials and Methods

### 3.1. Experimental Procedures for Phytochmistry Study

#### 3.1.1. General Experimental Procedures

The following instruments were used to obtain physical data—NMR spectra were measured on Bruker ascend 600 MHz or a Bruker ascend 500 MHz NMR spectrometer (Bruker BioSpin AG Industriestrasse 26 CH-8117, Fällanden, Switzerland) with tetramethylsilane as an internal standard. The negative-ion mode ESI-Q-Orbitrap MS was obtained on a Thermo UltiMate 3000 UHPLC instrument (Thermo, Waltham, MA, USA). Optical rotations, UV, and IR spectra were run on a Rudolph Autopol^®^ IV automatic polarimeter (l = 50 mm) (Rudolph Research Analytical, Hackettstown NJ, USA), Varian Cary 50 UV–Vis (Varian, Inc., Hubbardsdon, MA, USA) and Varian 640-IR FT-IR spectrophotometer (Varian Australia Pty Ltd., Mulgrave, Australia), respectively.

CC were performed on macroporous resin D101 (Haiguang Chemical Co., Ltd., Tianjin, China), silica gel (48–75μm, Qingdao Haiyang Chemical Co., Ltd., Qingdao, China), ODS (50 μm, YMC Co., Ltd., Tokyo, Japan), MCI gel (Mitsubishi Chemical Corporation, Osaka, Japan), and Sephadex LH-20 (Ge Healthcare Bio-Sciences, Uppsala, Sweden). High performance liquid chromatography (HPLC) column—Cosmosil 5C_18_-MS-II (4.6 mm i.d. × 250 mm, 5 µm) and Cosmosil 5C_18_-MS-II (20 mmi.d. × 250 mm, 5 µm, NakalaiTesque, Inc., Tokyo, Japan) were used to analyze and separate the constituents.

#### 3.1.2. Plant Material

The leaves of *Panax notoginseng* (Burk.) F. H. Chen were collected from Shilin country, Kunming city, Yunnan province, China, and identified by Dr. Wang Tao (Institute of Traditional Chinese Medicine, Tianjin University of Traditional Chinese Medicine). The voucher specimen was deposited at the Academy of Traditional Chinese Medicine of Tianjin University of TCM.

#### 3.1.3. Extraction and Isolation

The dried leaves of *P. notoginseng* (8 kg) was extracted three times with 50% EtOH under reflux for 3 h, 2 h, and 2 h, successively. Evaporation of the solvent under reduced pressure provided the 50% EtOH extract (2.67 kg). Then, an aliquot (2.1 kg) of the 50% EtOH extract was subjected to D101 resin CC (H_2_O → 95% EtOH) to give H_2_O- (760.0 g) and 95% EtOH eluted fraction (695.0 g), respectively.

The 95% EtOH eluate (150.0 g) was separated by silica gel CC [CH_2_Cl_2_→ CH_2_Cl_2_-MeOH (100:3 → 100:7 → 10:1 → 8:1 → 3:1 → 2:1 → 1:1, *v*/*v*)→ MeOH], and 12 fractions (Fr. 1–Fr. 12) were yielded. Fraction 6 (6.2 g) was further subjected to silica gel CC [CH_2_Cl_2_-MeOH-H_2_O (40:3:1 → 30:3:1 → 25:3:1 → 15:3:1 → 13:3:1 → 12:3:1 → 10:3:1 → 8:3:1 → 6:3:1, *v/v/v*, lower layer) → MeOH] to give 13 fractions (Fr. 6-1–Fr. 6-13). Fraction 6-5 (650.0 mg) was isolated by pHPLC [CH_3_CN-1% HAc (37:63, *v*/*v*)] to gain notoginsenoside NL-A_4_ (**4**, 8.1 mg). Fraction 6-10 (800.0 mg) was purified by pPHPLC [CH_3_CN-1% HAc (37:63, *v*/*v*)], and notoginsenosides NL-A_1_ (**1**, 44.9 mg), NL-C_1_ (**8**, 12.4 mg), as well as NL-D (**11**, 32.9 mg) were given. Fraction 7 (30.0 g) was separated by MCI gel CC [MeOH-H_2_O (65% → 70% → 75% → 80% → 100%, *v*/*v*)] to provide 12 fractions (Fr. 7-1–Fr. 7-12). Fraction 7-5 (480.0 mg) was further isolated by pHPLC [CH_3_CN-1% HAc (30:70, *v*/*v*)], and notoginsenoside NL-B_1_ (**5**, 23.8 mg) was yielded. Fraction 7-6 (800.0 mg) was subjected to pHPLC [MeOH-1% HAc (70:30, *v*/*v*)] to give seven fractions (Fr. 7-6-1–Fr. 7-6-7). Fraction 7-6-3 (141.5 mg) was purified by pHPLC [CH_3_CN-1% HAc (35:65, *v*/*v*)] to gain notoginsenoside NL-C_2_ (**9**, 11.9 mg). Fraction 7-6-4 (179.5 mg) was isolated by pHPLC [CH_3_CN-1% HAc (32:68, *v*/*v*)] to produce notoginsenoside NL-A_2_ (**2**, 8.0 mg). Fraction 8 (25.0 g) was subjected to MCI gel CC [MeOH-H_2_O (40% → 60% → 70% → 80% → 100%, *v*/*v*)], as a result, 11 fractions (Fr. 8-1–Fr. 8-11) were gained. Fraction 8-6 (1.6 g) was separated by pHPLC [MeOH-1% Hac (60:40, *v*/*v*)] to yield 13 fractions (Fr. 8-6-1–Fr. 8-6-13). Fraction 8-6-6 (135.3 mg) was further purified by pHPLC [CH_3_CN-1% HAc (26:74, *v*/*v*)] to obtain notoginsenosides NL-A_3_ (**3**, 31.2 mg) and NL-B_2_ (**6**, 10.2 mg). Fraction 8-6-9 (223.8 mg) was isolated by pHPLC [CH_3_CN-1% HAc (27:73, *v*/*v*)] to gain notoginsenoside NL-C_3_ (**10**, 37.4 mg). Fraction 9 (15.0 g) was subjected to MCI CC [MeOH-H_2_O (60% → 70% → 80% → 100%, *v*/*v*)], and 10 fractions (Fr. 9-1–Fr. 9-10) were given. Fraction 9-4 (500.0 mg) was separated by pHPLC [CH_3_CN-1% HAc (26:74, *v*/*v*)] to yield 11 fractions (Fr. 9-4-1–Fr. 9-4-10). Fraction 9-4-10 (10.1 mg) was further purified by pHPLC [MeOH-1% HAc (60:40, *v*/*v*)], and notoginsenoside NL-B_3_ (**7,** 5.6 mg) was gained.

*Notoginsenoside NL-A_1_* (**1**): White powder; ${\left[\alpha \right]}_{D}^{25}$ −1.8 (conc. 0.87, MeOH); IR (KBr) *ν*_max_3368, 2942, 2877, 1635, 1455, 1384, 1075, 1037, 895 cm^−1^; ^1^H-NMR (C_5_D_5_N, 500 MHz) and ^13^C-NMR (C_5_D_5_N, 125 MHz) spectroscopic data—see Table 1; ESI-Q-Orbitrap MS *m*/*z* 947.52405 [M − H]^−^ (calcd. for C_47_H_79_O_19_, 947.52101).

*Notoginsenoside NL-A_2_* (**2**): White powder; ${\left[\alpha \right]}_{D}^{25}$ +6.0 (conc. 0.26, MeOH); IR (KBr) *ν*_max_ 3418, 2968, 2865, 1647, 1455, 1395, 1052, 1033, 1014 cm^−1^; ^1^H-NMR (C_5_D_5_N, 500 MHz) and ^13^C-NMR (C_5_D_5_N, 125 MHz) spectroscopic data—see Table 2; ESI-Q-Orbitrap MS *m*/*z* 947.51996 [M − H]^−^ (calcd. for C_47_H_79_O_19_, 947.52101).

*Notoginsenoside NL-A_3_* (**3**): White powder; ${\left[\alpha \right]}_{D}^{25}$ −15.0 (conc. 0.80, MeOH); IR (KBr) ν_max_ 3368, 2939, 2877, 1635, 1453, 1386, 1074, 1042 cm^−1^; ^1^H-NMR (C_5_D_5_N, 500 MHz)and ^13^C-NMR (C_5_D_5_N, 125 MHz) spectroscopic data—see Table 3; ESI-Q-Orbitrap MS *m*/*z* 1241.61584 [M − H]^−^ (calcd. for C_58_H_97_O_28_, 1241.61609).

*Notoginsenoside NL-A_4_* (**4**): White powder; ${\left[\alpha \right]}_{D}^{25}$ +18.7 (conc. 0.31, MeOH); IR (KBr) *ν*_max_ 3379, 2946, 2877, 1718, 1646, 1456, 1384, 1080, 1041, 1010 cm^−1^; ^1^H-NMR (C_5_D_5_N, 500 MHz) and ^13^C-NMR (C_5_D_5_N, 125 MHz) spectroscopic data—see Table 4; ESI-Q-Orbitrap MS *m*/*z* 871.46875 [M − H]^−^ (calcd. for C_44_H_71_O_17_, 871.46858).

*Notoginsenoside NL-B_1_* (**5**): White powder; ${\left[\alpha \right]}_{D}^{25}$ +2.2 (conc. 1.19, MeOH); IR (KBr) *ν*_max_ 3379, 2944, 2877, 1649, 1458, 1387, 1076, 1035, 1018 cm^−1^; ^1^H-NMR (C_5_D_5_N, 500 MHz) and ^13^C-NMR (C_5_D_5_N, 125 MHz) spectroscopic data—see Table 5; ESI-Q-Orbitrap MS *m*/*z* 931.52838 [M − H]^−^ (calcd. for C_47_H_79_O_18_, 931.52609).

*Notoginsenoside NL-B_2_* (**6**): White powder; ${\left[\alpha \right]}_{D}^{25}$ −2.1 (conc. 0.28, MeOH); IR (KBr) *ν*_max_ 3364, 2938, 2877, 1647, 1456, 1384, 1073, 1042, 893 cm^−1^; ^1^H-NMR (C_5_D_5_N, 500 MHz)and ^13^C-NMR (C_5_D_5_N, 125 MHz) spectroscopic data—see Table 6; ESI-Q-Orbitrap MS *m*/*z* 1225.62134 [M − H]^−^ (calcd. for C_58_H_97_O_27_, 1225.62117).

*Notoginsenoside NL-B_3_* (**7**): White powder; ${\left[\alpha \right]}_{D}^{25}$ −1.7 (conc. 0.23, MeOH); IR (KBr) *ν*_max_ 3360, 2927, 2877, 1630, 1455, 1382, 1073, 1041, 894 cm^−1^; ^1^H-NMR (C_5_D_5_N, 600 MHz) and ^13^C-NMR (C_5_D_5_N, 150 MHz) spectroscopic data—see Table 7; ESI-Q-Orbitrap MS *m*/*z* 1225.61841 [M − H]^−^ (calcd. for C_58_H_97_O_27_, 1225.62117).

*Notoginsenoside NL-C_1_* (**8**): White powder; ${\left[\alpha \right]}_{D}^{25}$ −6.7 (conc. 0.21, MeOH); UV *λ*_max_ (MeOH) nm (log *ε*) 218 (3.57); IR (KBr) *ν*_max_ 3387, 2941, 2877, 1669, 1632, 1456, 1384, 1077, 1039 cm^−1^; ^1^H-NMR (C_5_D_5_N, 500 MHz)and ^13^C-NMR (C_5_D_5_N, 125 MHz) spectroscopic data—see Table 8; ESI-Q-Orbitrap MS *m*/*z* 929.51184 [M − H]^−^ (calcd. for C_47_H_77_O_18_, 929.51044).

*Notoginsenoside NL-C_2_* (**9**): White powder; ${\left[\alpha \right]}_{D}^{25}$ +1.8 (conc. 0.45, MeOH); UV *λ*_max_ (MeOH) nm (log *ε*) 219 (3.63); IR (KBr) *ν*_max_ 3415, 2943, 2877, 1671, 1639, 1458, 1386, 1078, 1038, 894 cm^−1^; ^1^H-NMR (C_5_D_5_N, 500 MHz) and ^13^C-NMR (C_5_D_5_N, 125 MHz) spectroscopic data—see Table 9; ESI-Q-Orbitrap MS *m*/*z* 929.51221 [M − H]^−^ (calcd. for C_47_H_77_O_18_, 929.51044).

*Notoginsenoside NL-C_3_* (**10**): White powder; ${\left[\alpha \right]}_{D}^{25}$ +1.6 (conc. 0.51, MeOH); UV *λ*_max_ (MeOH) nm (log *ε*) 218 (3.69); IR (KBr) *ν*_max_ 3391, 2942, 2881, 1663, 1645, 1453, 1385, 1164, 1077, 1040, 895 cm^−1^; ^1^H-NMR (C_5_D_5_N, 500 MHz) and ^13^C-NMR (C_5_D_5_N, 125 MHz) spectroscopic data—see Table 10; ESI-Q-Orbitrap MS *m*/*z* 1091.56445 [M − H]^−^ (calcd. for C_53_H_87_O_23_, 1091.56327).

*Notoginsenoside NL-D* (**11**): White powder; ${\left[\alpha \right]}_{D}^{25}$ +29.6 (conc. 0.34, MeOH); IR (KBr) *ν*_max_ 3382, 2953, 2878, 1717, 1649, 1456, 1387, 1312, 1075, 1048, 1008 cm^−1^; ^1^H-NMR (C_5_D_5_N, 500 MHz) and ^13^C-NMR (C_5_D_5_N, 125 MHz) spectroscopic data—see Table 11; ESI-Q-Orbitrap MS *m*/*z* 855.47388 [M − H]^−^ (calcd. for C_44_H_71_O_16_, 855.47366).

Acid Hydrolysis of **1**–**11**: Compounds **1**–**11** were hydrolyzed by acid and an aqueous layer was obtained by using the method reported in [12]—their solution (each 2.0 mg) in 1 M HCl (1.0 mL) was heated under reflux for 3 h, respectively. Then, each reaction mixture was neutralized with Amberlite IRA-400 (OH− form) and removed by filtration. Next, the aqueous layer was subjected to the HPLC analysis under the following conditions—HPLC column, Kaseisorb LC NH_2_-60-5, 4.6 mm i.d. × 250 mm (Tokyo Kasei Co. Ltd., Tokyo, Japan); detection and optical rotation [Chiralyser (IBZ Messtechnik GMBH, Mozartstrasse 14-16 D-30173 Hannover, Germany)]; a mobile phase, CH_3_CN-H_2_O (80:20, *v*/*v*); and a flow rate of 0.7 mL/min. Then, d-xylose (from **2**, **3**, 6, **7**, **9**, and **10**), l-arabinose (from **1**, **3**–**6**, **8**, and **11**), and d-glucose (from **1**–**11**) and were confirmed by comparison of the retention times with the authentic samples [*t*_R_: 9.4 min (d-xylose), 10.3 min (l-arabinose), and 12.4 min (d-glucose), all of them showed positive optical rotations].

### 3.2. Experimental Procedures for Bioassay

The MTT and nitrite levels measurements of **1**–**11**, as well as statistical analyses were conducted by using the method reported previously [16].

## 4. Conclusions

It is well-known that PNS are the major bioactive saponins in *P. notoginseng*. They were chemically classified as the derivatives of 20(*S*)-protopanaxatriol (PPT) and 20(S)-protopanaxadiol (PPD). In the present study, eleven new dammarane-type triterpenoid saponins were obtained and identified from the leaves extract of *P. notoginseng*. Among them, **1**–**10** were PPD-type saponins. It was suggested that PPD derivatives were the major PNS in *P. notoginseng* leaves, which was in accordance with the results found by other researchers [6,7,8,9]. As PPTs were reported to be the main PNS in *P. notoginseng* roots [17], we could not draw the conclusion that *P. notoginseng* leaves would be a possible replacement of the roots used in clinics.

From a structural point of view, the aglycones of compounds **1**–**10** were dammarane-type with oxygenated C-17 side chains different to protopanaxadiol. Compounds **1**–**4**, **5**–**7**, and **8**–**10** featured 23-ene25-hydroperoxyl, 23-ene-25-hydroxyl, and α,β-unsaturated ketone, respectively. Among them, 23-ene-25-hydroperoxyl-substituted protopanaxadiol saponins were found in *P. notoginseng* leaves, first though they were found in *P. notoginseng* flowers [18], roots [19], and rhizomes [20]. Additionally, compounds **4** and **11** were characterized by the malonyl substitution at 3-position. The 3-malonyl substituted dammarane-type terpennoids are reported here for the first time. These results enrich the chemical investigations of *P. notoginseng* leaves.

Inflammation is widespread in the clinical pathology and closely associated to the progress of many diseases. NO is implicated in a variety of inflammatory conditions, which indicates the agents block NO production might be beneficial for the treatment of inflammatory responses. In this study, pretreatment of notoginsenosides NL-A_1_ (**1**), NL-A_2_ (**2**), NL-B_1_ (**5**), NL-B_2_ (**6**), NL-C_1_–NL-C_3_ (**8**–**10**) at noncytotoxic concentration (50 μM) decreased the NO production significantly, which indicated that they have an anti-inflammatory activity. Moreover, the inhibitory effects of all of them were found to be dose-dependent. Further investigations on their anti-inflammatory mechanisms would be beneficial.

## Figures and Tables

**Figure 1 molecules-25-00139-f001:**
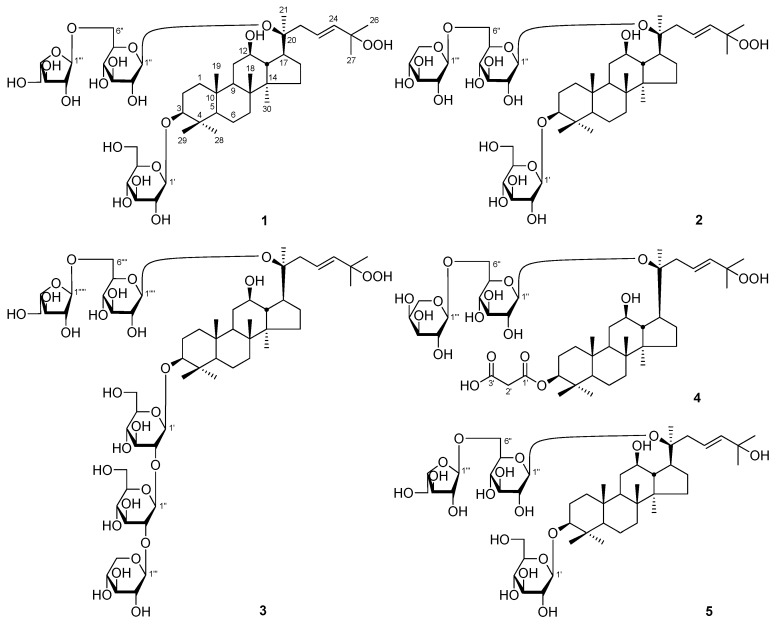
The new compounds **1**–**11** obtained from *Panax notoginseng* leaves.

**Figure 2 molecules-25-00139-f002:**
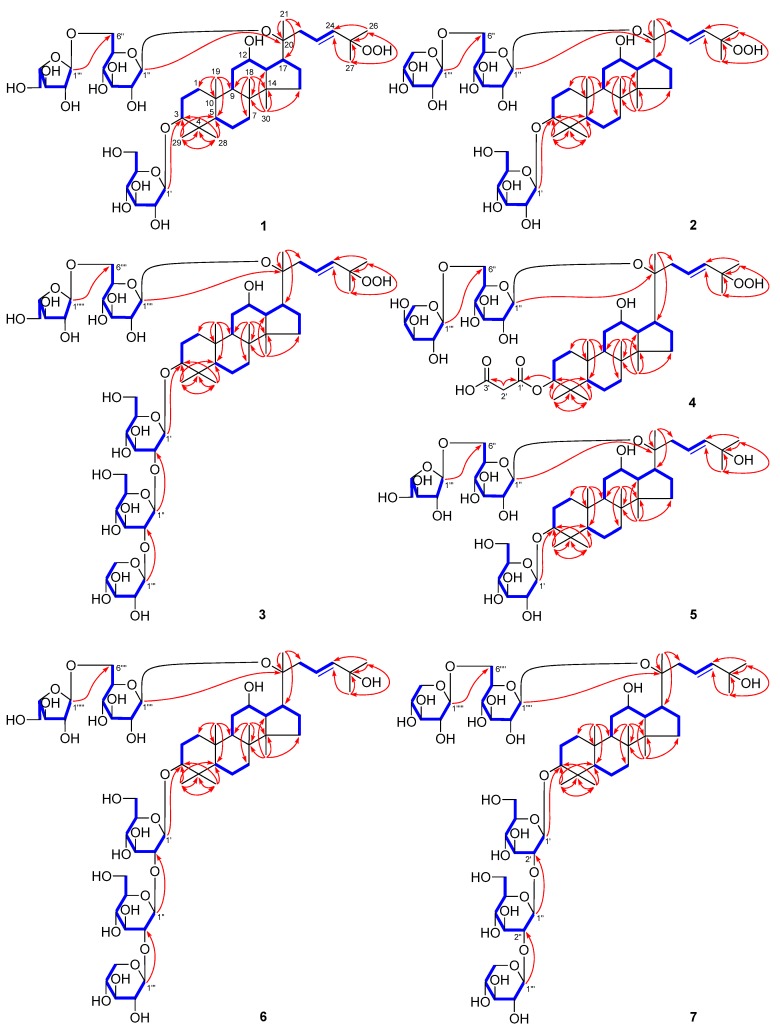
Key ^1^H–^1^H COSY and HMBC correlations of **1**–**11**.

**Figure 3 molecules-25-00139-f003:**
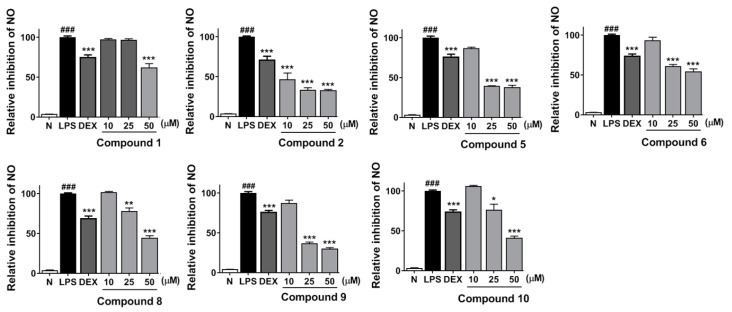
Inhibitory effects of compounds **1**, **2**, **5**, **6**, **8**–**10** at concentrations of 10, 25, and 50 μM on NO production in RAW 264.7 cells, respectively. Nitrite relative concentration (NRC)—percentage of control group (set as 100%). Values represent the mean ± SD of four determinations. * *p* < 0.05, ** *p* < 0.01, *** *p* < 0.001 vs. LPS group; ### *p* < 0.001 vs normal group (*N*) Differences between compound-treated group and control group; *N* = 4.

**Table 1 molecules-25-00139-t001:** The ^1^H and ^13^C-NMR data for **1** in C_5_D_5_N.

No.	δ_C_	δ_H_ (*J* in Hz)	No.	δ_C_	δ_H_ (*J* in Hz)
1	39.1	0.78, 1.56 (both m)	25	81.3	-
2	26.7	1.81 (m, overlapped)	26	25.4	1.61 (s)
		2.21 (m)			
3	88.8	3.36 (dd, 4.0, 11.5)	27	25.1	1.61 (s)
4	39.7	-	28	28.1	1.32 (s)
5	56.4	0.73 (br. d, *ca.* 12)	29	16.8	1.00 (s)
6	18.4	1.39 (m, overlapped)	30	17.2	0.90 (s)
		1.51 (m)			
7	35.0	1.22, 1.49 (both m)	1′	107.1	4.95 (d, 8.0)
8	40.0	-	2′	75.7	4.06 (m, overlapped)
9	50.1	1.39 (m, overlapped)	3′	78.7	4.27 (dd, 8.0, 9.0)
10	36.9	-	4′	71.8	4.22 (m)
11	30.7	1.56 (m)	5′	78.4	4.03 (m, overlapped)
		2.01 (m, overlapped)			
12	70.6	4.02 (m)	6′	63.0	4.42 (dd, 5.5, 11.5)
					4.62 (dd, 2.0, 11.5)
13	49.4	2.01 (m, overlapped)	1″	98.2	5.18 (d, 8.0)
14	51.5	-	2″	75.1	3.97 (dd, 7.5, 8.0),
15	30.5	0.98 (m)	3″	78.8	4.20 (m, overlapped)
		1.61 (m, overlapped)			
16	26.3	1.47, 1.79 (both m)	4″	71.9	4.03 (m, overlapped)
17	52.0	2.44 (q like, *ca.* 11)	5″	76.4	4.09 (m, H-5″)
18	16.0	1.02 (s)	6″	68.3	4.13 (dd, 5.0, 11.0)
					4.66 (br. d, *ca.* 11)
19	16.3	0.81 (s)	1‴	110.0	5.66 (d, 1.5)
20	83.2	-	2‴	83.3	4.87 (br. s),
21	23.3	1.61 (s)	3‴	78.9	4.78 (m)
22	39.8	2.81 (dd, 8.0, 14.0)	4‴	85.9	4.78 (m)
		3.08 (dd, 5.5, 14.0)			
23	126.7	6.16 (ddd, 5.5, 8.0, 16.0)	5‴	62.7	4.21 (m, overlapped)
					4.31 (br. d, *ca.* 11)
24	138.0	6.11 (d, 16.0)			

**Table 2 molecules-25-00139-t002:** The ^1^H and ^13^C-NMR data for **2** in C_5_D_5_N.

No.	δ_C_	δ_H_ (*J* in Hz)	No.	δ_C_	δ_H_ (*J* in Hz)
1	39.1	0.78 (m)	25	81.4	-
		1.57 (m, overlapped)			
2	26.8	1.84, 2.23 (both m)	26	25.5	1.61 (s)
3	88.8	3.38 (dd, 4.5, 11.5)	27	25.1	1.62 (s)
4	39.7	-	28	28.2	1.33 (s)
5	56.4	0.74 (br. d, *ca.* 12)	29	16.8	1.02 (s)
6	18.5	1.39, 1.54 (both m)	30	17.2	0.93 (s)
7	35.1	1.22, 1.50 (both m)	1′	107.1	4.98 (d, 7.5)
8	40.1	-	2′	75.8	4.08 (dd, 7.5, 8.5)
9	50.1	1.42 (m)	3′	78.8	4.29 (dd, 8.5, 9.0)
10	37.0	-	4′	71.9	4.26 (dd, 9.0, 9.0)
11	30.9	1.55 (m, overlapped)	5′	78.5	4.06 (m, overlapped)
		2.02 (m)			
12	70.4	4.08 (m)	6′	63.1	4.44 (dd, 5.5, 11.5)
					4.64 (br. d, *ca.* 12)
13	49.7	2.04 (dd, 10.5, 10.5)	1″	98.3	5.20 (d, 7.5)
14	51.5	-	2″	75.0	3.95 (dd, 7.5, 8.5)
15	30.6	0.99 (m)	3″	79.0	4.21 (m, overlapped)
		1.60 (m, overlapped)			
16	26.4	1.47, 1.80 (both m)	4″	71.6	4.13 (m, overlapped)
17	51.9	2.49 (q like, *ca.* 11)	5″	76.9	4.13 (m, overlapped)
18	16.0	1.02 (s)	6″	69.9	4.33 (m, overlapped)
					4.76 (br. d, *ca.* 11)
19	16.3	0.85 (s)	1‴	105.7	5.00 (d, 7.5)
20	83.2	-	2‴	74.9	4.05 (dd, 7.5, 8.0)
21	23.3	1.62 (s)	3‴	78.1	4.15 (dd, 8.0, 9.0)
22	39.7	2.85 (dd, 8.0, 13.5)	4‴	71.1	4.22 (m)
		3.14 (dd, 5.5, 13.5)			
23	126.8	6.21 (ddd, 5.5, 8.0, 16.0)	5‴	67.1	3.74 (dd, 10.5, 10.5)
					4.34 (m, overlapped)
24	138.1	6.13 (d, 16.0)			

**Table 3 molecules-25-00139-t003:** The ^1^H and ^13^C-NMR data for **3** in C_5_D_5_N.

No.	δ_C_	δ_H_ (*J* in Hz)	No.	δ_C_	δ_H_ (*J* in Hz)
1	39.2	0.76 (m)	30	17.2	0.88 (s)
		1.54 (m, overlapped)			
2	26.8	1.81, 2.15 (both m)	1′	104.8	4.92 (d, 7.5)
3	89.0	3.28 (dd, 4.5, 11.5),	2′	82.9	4.09 (m, overlapped)
4	39.9	-	3′	78.7	4.33 (m, overlapped)
5	56.4	0.69 (br. d, *ca.* 12)	4′	71.2	4.09 (m, overlapped)
6	18.5	1.46 (m, overlapped)	5′	78.3	3.94 (m)
		1.53 (m)			
7	35.1	1.20 (m)	6′	63.0	4.35 (m, overlapped)
		1.46 (m, overlapped)			4.56 (dd, 2.0, 12.0)
8	40.1	-	1″	103.2	5.50 (d, 7.5)
9	50.1	1.36 (m)	2″	84.6	4.18 (m, overlapped)
10	36.9	-	3″	77.9	4.28 (dd, 8.5, 9.5)
11	30.8	1.55 (m, overlapped)	4″	71.8	4.19 (m, overlapped)
		1.98 (m)			
12	70.7	3.99 (m)	5″	77.7	3.85 (m)
13	49.5	2.02 (dd, 10.5, 10.5)	6″	62.9	4.35 (m, overlapped)
					4.46 (dd, 2.5, 11.5)
14	51.5	-	1‴	106.4	5.39 (d, 7.0)
15	30.7	0.97 (m)	2‴	75.9	4.09 (m, overlapped)
		1.54 (m, overlapped)			
16	26.4	1.46 (m, overlapped)	3‴	77.8	4.12 (m, overlapped)
		1.79 (m)			
17	52.1	2.43 (q like, *ca.* 11)	4‴	70.7	4.12 (m, overlapped)
18	16.0	1.01 (s)	5‴	67.4	3.68 (dd, 10.0, 11.0)
					4.32 (m, overlapped)
19	16.3	0.83 (s)	1⁗	98.2	5.17 (d, 7.5)
20	83.2	-	2⁗	75.2	3.95 (dd, 7.5, 9.0)
21	23.3	1.61 (s)	3⁗	78.9	4.18 (dd, 9.0, 9.0)
22	39.8	2.81 (dd, 8.0, 14.5)	4⁗	72.0	4.03 (dd, 9.0, 9.0)
		3.10 (dd, 6.0, 14.5)			
23	126.7	6.14 (ddd, 6.0, 8.0, 16.0)	5⁗	76.5	4.04 (m)
24	138.0	6.10 (d, 16.0)	6⁗	68.4	4.12 (m, overlapped)
					4.65 (dd, 1.5, 11.5)
25	81.3	-	1′′′′′	110.2	5.64 (d, 1.5)
26	25.4	1.60 (s)	2′′′′′	83.3	4.85 (dd, 1.5, 3.0)
27	25.1	1.60 (s)	3′′′′′	78.8	4.77 (m, overlapped)
28	28.1	1.28 (s)	4′′′′′	86.0	4.77 (m, overlapped)
29	16.7	1.11 (s)	5′′′′′	62.8	4.19 (m, overlapped)
					4.29 (dd, 3.0, 12.0)

**Table 4 molecules-25-00139-t004:** The ^1^H and ^13^C-NMR data for **4** in C_5_D_5_N.

No.	δ_C_	δ_H_ (*J* in Hz)	No.	δ_C_	δ_H_ (*J* in Hz)
1	38.3	0.81 (m, overlapped)	23	126.6	6.15 (m)
		1.59 (m, overlapped)			
2	24.0	1.70, 1.76 (both m)	24	138.1	6.15 (m)
3	81.3	4.81 (dd, 4.5, 11.5)	25	81.3	-
4	38.6	-	26	25.4	1.62 (s)
5	56.1	0.78 (br. d, *ca.* 11)	27	25.1	1.61 (s)
6	18.4	1.38 (m, overlapped)	28	28.1	1.05 (s)
		1.47 (m, overlapped)			
7	34.9	1.19 (m)	29	16.8	0.94 (s)
		1.46 (m, overlapped)			
8	40.0	-	30	17.2	0.87 (s)
9	50.1	1.38 (m, overlapped)	1′	168.3	-
10	37.1	-	2′	43.8	3.83 (s)
11	30.8	1.58 (m, overlapped)	3′	169.9	-
		1.99 (m)			
12	70.4	4.01 (m)	1″	98.3	5.20 (d, 7.0)
13	49.5	2.03 (dd, 10.0, 10.0)	2″	75.1	3.96 (dd, 7.0, 8.0)
14	51.4	-	3″	78.8	4.21 (dd, 8.0, 8.5)
15	30.5	0.98 (m, overlapped)	4″	71.7	4.14 (dd, 8.5, 9.0)
		1.58 (m, overlapped)			
16	26.4	1.49, 1.80 (both m)	5″	76.7	4.10 (m)
17	52.0	2.45 (q like, *ca.* 10)	6″	69.0	4.28 (dd, 4.0, 10.5)
					4.71 (br. d, *ca.* 11)
18	16.0	1.02 (s)	1‴	104.4	5.02 (d, 6.0)
19	16.2	0.84 (s)	2‴	72.1	4.47 (dd, 6.0, 6.5)
20	83.3	-	3‴	74.1	4.25 (m)
21	23.3	1.62 (s)	4‴	68.5	4.38 (m)
22	40.1	2.88 (dd, 6.5, 13.5)	5‴	65.4	3.85 (br. d, *ca.* 12)
		3.10 (dd, 4.0, 13.5)			4.34 (dd, 2.5, 12.0)

**Table 5 molecules-25-00139-t005:** The ^1^H and ^13^C-NMR data for **5** in C_5_D_5_N.

No.	δ_C_	δ_H_ (*J* in Hz)	No.	δ_C_	δ_H_ (*J* in Hz)
1	39.1	0.78 (m)	25	70.0	-
		1.55 (m, overlapped)			
2	26.8	1.79 (m, overlapped)	26	30.7	1.56 (s)
		2.22 (m)			
3	89.0	3.37 (dd, 4.5, 12.0)	27	30.6	1.57 (s)
4	39.7	-	28	28.1	1.28 (s)
5	56.4	0.73 (br. d, *ca.* 12)	29	16.7	1.11 (s)
6	18.4	1.38 (m, overlapped)	30	17.2	0.89 (s)
		1.50 (m)			
7	35.1	1.20 (m)	1′	107.0	4.96 (d, 8.0)
		1.48 (m, overlapped)			
8	40.1	-	2′	75.8	4.06 (dd, 8.0, 9.0)
9	50.1	1.37 (m, overlapped)	3′	78.8	4.28 (dd, 9.0, 9.0)
10	36.9	-	4′	71.8	4.24 (dd, 9.0, 9.0)
11	30.8	1.56 (m, overlapped)	5′	78.4	4.05 (m, overlapped)
		2.04 (m)			
12	70.6	4.04 (m)	6′	63.1	4.44 (dd, *J* = 5.5, 11.5)
					4.63 (dd, 1.5, 11.5)
13	49.5	2.04 (dd, 11.0, 11.0)	1″	98.2	5.18 (d, 7.5)
14	51.5	-	2″	75.2	3.98 (dd, 7.5, 8.5)
15	30.5	0.96 (m)	3″	78.9	4.18 (dd, 8.5, 8.5)
		1.56 (m, overlapped)			
16	26.4	1.47, 1.79 (both m, overlapped)	4″	72.0	4.01 (dd, 8.5, 9.0)
17	52.1	2.44 (q like, *ca.* 10)	5″	76.6	4.09 (m)
18	16.0	1.01 (s)	6″	68.4	4.21 (dd, 3.0, 11.0)
					4.69 (br. d, *ca.* 11)
19	16.3	0.83 (s)	1‴	110.2	5.65 (d, 1.5)
20	83.3	-	2‴	83.5	4.89 (br. s)
21	23.3	1.61 (s)	3‴	78.9	4.80 (m, overlapped)
22	39.6	2.84 (dd, 8.5, 14.0)	4‴	86.0	4.80 (m, overlapped)
		3.11 (dd, 5.5, 14.0)			
23	122.7	6.23 (ddd, 5.5, 8.5, 15.5)	5‴	62.8	4.21 (dd, 3.5, 12.0)
					4.32 (br. d, *ca.* 12)
24	142.2	6.09 (d, 15.5)			

**Table 6 molecules-25-00139-t006:** The ^1^H and ^13^C-NMR data for **6** in C_5_D_5_N.

No.	δ_C_	δ_H_ (*J* in Hz)	No.	δ_C_	δ_H_ (*J* in Hz)
1	39.2	0.76, 1.52 (both m)	30	17.2	0.89 (s)
2	26.8	1.81, 2.18 (both m)	1′	104.8	4.92 (d, 8.0)
3	89.0	3.28 (dd, 4.0, 11.5)	2′	83.0	4.11 (dd, 7.0, 8.0)
4	39.8	-	3′	78.7	4.35 (m, overlapped)
5	56.4	0.69 (br. d, *ca.* 12)	4′	71.2	4.08 (dd, 9.0, 9.0)
6	18.5	1.37 (m, overlapped)	5′	78.3	3.94 (m)
		1.49 (m)			
7	35.1	1.20, 1.45 (both m)	6′	63.0	4.35 (m, overlapped)
					4.57 (dd, 2.0, 11.5)
8	40.1	-	1″	103.2	5.50 (d, 7.5)
9	50.1	1.37 (m, overlapped)	2″	84.6	4.19 (m, overlapped)
10	36.9	-	3″	78.0	4.27 (dd, *J* = 9.0, 9.0)
11	30.8	1.55 (m, overlapped)	4″	71.9	4.20 (dd, 7.0, 9.0)
		1.98 (m)			
12	70.6	3.99 (m, overlapped)	5″	77.8	3.85 (m)
13	49.5	2.04 (dd, 10.5, 10.5)	6″	62.9	4.35 (m, overlapped)
					4.46 (dd, 3.0, 11.5)
14	51.5	-	1‴	106.5	5.39 (d, 6.5)
15	30.6	0.96 (m)	2‴	76.0	4.10 (dd, 6.5, 9.0)
		1.55 (m, overlapped)			
16	26.4	1.47, 1.77 (both m)	3‴	77.7	4.12 (m, overlapped)
17	52.1	2.44 (q like, *ca.* 11)	4‴	70.7	4.13 (dd, 9.0, 9.0)
18	16.0	1.01 (s)	5‴	67.4	3.68 (dd, 11.0, 11.0)
					4.30 (m, overlapped)
19	16.3	0.83 (s)	1⁗	98.2	5.16 (d, 7.5)
20	83.4	-	2⁗	75.2	3.95 (dd, 7.5, 8.0)
21	23.3	1.60 (s)	3⁗	78.9	4.16 (dd, 8.0, 9.5)
22	39.6	2.82 (dd, 8.5, 14.0)	4⁗	72.0	3.98 (dd, 9.0, 9.5)
		3.09 (dd, 6.0, 14.0)			
23	122.8	6.21 (ddd, 6.0, 8.5, 15.5)	5⁗	76.5	4.06 (m)
24	142.3	6.07 (d, 15.5)	6⁗	68.4	4.07 (m, overlapped)
					4.65 (br. d, *ca.* 11)
25	70.0	-	1′′′′′	110.2	5.64 (d, 1.5)
26	30.7	1.55 (s)	2′′′′′	83.3	4.85 (dd, 1.5, 3.0)
27	30.7	1.55 (s)	3′′′′′	79.0	4.77 (m, overlapped)
28	28.1	1.28 (s)	4′′′′′	86.1	4.77 (m, overlapped)
29	16.7	1.11 (s)	5′′′′′	62.8	4.19 (m, overlapped)
					4.30 (m, overlapped)

**Table 7 molecules-25-00139-t007:** The ^1^H and ^13^C-NMR data for 7 in C_5_D_5_N

No.	δ_C_	δ_H_ (*J* in Hz)	No.	δ_C_	δ_H_ (*J* in Hz)
1	39.2	0.76, 1.53 (both m)	30	17.2	0.91 (s)
2	26.8	1.80, 2.19 (both m)	1′	104.8	4.94 (d, 7.2)
3	89.0	3.30 (dd, 4.2, 11.4)	2′	82.8	4.14 (m, overlapped)
4	39.8	-	3′	78.7	4.37 (m, overlapped)
5	56.4	0.69 (br. d, *ca.* 11)	4′	71.1	4.10 (m, overlapped)
6	18.5	1.37 (m, overlapped)	5′	78.3	3.98 (m)
		1.47 (m)			
7	35.1	1.21 (m)	6′	63.0	4.37 (m, overlapped)
		1.47 (m, overlapped)			4.59 (dd, 1.8, 11.4)
8	40.0	-	1″	103.1	5.54 (d, 7.8)
9	50.1	1.37 (m, overlapped)	2″	84.6	4.21 (dd, 7.8, 9.0)
10	36.9	-	3″	78.0	4.30 (dd, 9.0, 9.0)
11	30.7	1.55 (m, overlapped)	4″	71.8	4.22 (m, overlapped)
		1.99 (m)			
12	70.5	4.05 (m)	5″	77.8	3.88 (m)
13	49.6	2.05 (dd, 10.8, 10.8)	6″	62.9	4.37 (m, overlapped)
					4.51 (dd, 2.4, 11.4)
14	51.5	-	1‴	106.5	5.41 (d, 7.2)
15	30.6	0.97 (m)	2‴	76.0	4.12 (dd, 7.2, 9.6)
		1.55 (m, overlapped)			
16	26.4	1.49 (m, overlapped)	3‴	77.9	4.15 (m, overlapped)
		1.77 (m)			
17	52.0	2.47 (q like, *ca.* 10)	4‴	70.7	4.16 (m, overlapped)
18	16.0	1.00 (s)	5‴	67.4	3.71 (dd, 11.4, 11.4)
					4.36 (m, overlapped)
19	16.3	0.83 (s)	1⁗	98.2	5.18 (d, 7.8)
20	83.3	-	2⁗	75.1	3.95 (dd, 7.8, 8.4)
21	23.2	1.61 (s)	3⁗	78.9	4.19 (dd, 8.4, 9.0)
22	39.8	2.86 (dd, 8.4, 13.8)	4⁗	71.6	4.10 (m, overlapped)
		3.13 (dd, 6.0, 13.8)			
23	122.8	6.24 (ddd, 6.0, 8.4, 15.6)	5⁗	76.9	4.10 (m, overlapped)
24	142.3	6.10 (d, 15.6)	6⁗	70.0	4.28 (dd, 4.2, 13.2)
					4.76 (dd, 1.2, 13.2)
25	70.1	-	1′′′′′	105.6	4.98 (d, 7.2)
26	30.7	1.57 (s)	2′′′′′	74.9	4.04 (dd, 7.2, 8.4)
27	30.6	1.56 (s)	3′′′′′	78.0	4.20 (m, overlapped)
28	28.1	1.29 (s)	4′′′′′	71.1	4.17 (m)
29	16.7	1.12 (s)	5′′′′′	67.0	3.70 (dd, 9.6, 9.6)
					4.33 (dd, 4.8, 9.6)

**Table 8 molecules-25-00139-t008:** The ^1^H and ^13^C-NMR data for **8** in C_5_D_5_N

No.	δ_C_	δ_H_ (*J* in Hz)	No.	δ_C_	δ_H_ (*J* in Hz)
1	39.2	0.77, 1.56 (both m)	25	144.4	-
2	26.8	1.82(m, overlapped)	26	125.5	5.78, 6.37 (both br. s)
		2.23 (m)			
3	89.0	3.37 (dd, 2.5, 10.5 Hz)	27	17.8	1.85 (s)
4	39.7	-	28	28.2	1.31 (s)
5	56.4	0.72 (br. d, *ca.* 12)	29	16.8	1.00 (s)
6	18.5	1.38, 1.50 (both m, overlapped)	30	17.4	0.98 (s)
7	35.1	1.19, 1.47 (both m)	1′	107.0	4.95 (d, 7.5 Hz)
8	40.0	-	2′	75.8	4.06 (dd, 7.5, 8.0)
9	50.2	1.37 (m, overlapped)	3′	78.8	4.27 (dd, 8.0, 9.0)
10	37.0	-	4′	71.9	4.23 (dd, 8.5, 9.0)
11	30.8	1.51, 1.97 (both m, overlapped)	5′	78.4	4.03 (m)
12	70.2	4.21 (m)	6′	63.1	4.40 (dd, 5.5, 11.5)
					4.61 (dd, 1.5, 11.5)
13	49.5	2.04 (m)	1″	98.0	5.10 (d, 8.0)
14	51.5	-	2″	74.9	3.95 (dd, 8.0, 8.5)
15	30.7	0.99, 1.53 (both m, overlapped)	3″	79.4	4.19 (dd, 8.5, 9.0)
16	26.7	1.36, 1.82 (both m, overlapped)	4″	72.0	3.97 (dd, 9.0, 9.0)
17	51.9	2.60 (q like, *ca.* 10)	5″	76.7	4.09 (m)
18	16.0	0.92 (s)	6″	68.8	4.13 (dd, 5.0, 11.0)
					4.70 (br. d, *ca.* 11)
19	16.3	0.80 (s)	1‴	110.3	5.71 (d, 1.5)
20	83.2	-	2‴	83.5	4.88 (dd, 1.5, 4.5)
21	21.9	1.58 (s)	3‴	78.8	4.82 (dd, 4.5, 4.5)
22	29.8	2.09, 2.75 (both m)	4‴	86.0	4.76 (m)
					4.25 (m, overlapped)
23	32.7	3.17 (ddd, 6.5, 10.5, 16.5)	5‴	62.7	4.35 (dd, 3.0, 12.0)
		3.40 (ddd, 3.5, 10.5, 16.5)			
24	202.6	-			

**Table 9 molecules-25-00139-t009:** The ^1^H and ^13^C-NMR data for **9** in C_5_D_5_N

No.	δ_C_	δ_H_ (*J* in Hz)	No.	δ_C_	δ_H_ (*J* in Hz)
1	39.2	0.80 (m, overlapped)	25	144.4	-
		1.58 (m, overlapped)			
2	26.8	1.84 (m, overlapped)	26	125.5	5.72, 6.37 (both br. s)
		2.24 (m)			
3	88.9	3.38 (dd, 4.5, 11.5)	27	17.9	1.84 (s)
4	39.7	-	28	28.2	1.31 (s)
5	56.4	0.73 (br. d, *ca.* 12)	29	16.8	1.00 (s)
6	18.5	1.39(m, overlapped)	30	17.4	0.98 (s)
		1.51 (m, overlapped)			
7	35.1	1.20, 1.48 (both m)	1′	107.0	4.95 (d, 8.0 Hz)
8	40.1	-	2′	75.8	4.05 (dd, 8.0, 8.5)
9	50.3	1.37 (m, overlapped)	3′	78.8	4.26 (dd, 8.5, 8.5)
10	37.0	-	4′	71.9	4.22 (dd, 8.5, 9.0)
11	30.8	1.50 (m, overlapped)	5′	78.4	4.02 (m, overlapped)
		1.96 (m)			
12	70.1	4.21 (m)	6′	63.1	4.40 (dd, 5.5, 11.0)
					4.61 (dd, 2.0, 11.5)
13	49.5	2.03 (dd, 9.5, 9.5)	1″	98.0	5.10 (d, 7.5)
14	51.5	-	2″	74.8	3.90 (dd, 7.5, 8.0)
15	30.7	1.00, 1.58 (both m, overlapped)	3″	79.4	4.17 (dd, 8.0, 8.5)
16	26.7	1.37(m, overlapped)	4″	71.3	4.12 (dd, 8.5, 8.5)
		1.84 (m, overlapped)			
17	52.1	2.60 (q like, *ca.* 10 Hz)	5″	76.6	4.04 (m, overlapped)
18	16.0	0.93 (s)	6″	70.1	4.28 (dd, 5.5, 11.5)
					4.76 (dd, 2.0, 11.5)
19	16.3	0.82 (s)	1‴	106.1	4.91 (d, 7.5)
20	83.3	-	2‴	74.9	4.02 (m, overlapped)
21	21.7	1.58 (s)	3‴	78.2	4.13 (dd, 8.5, 8.5)
22	29.8	2.11 (m)	4‴	71.2	4.25 (m)
		2.75 (ddd, 5.0, 10.5, 14.5)			
23	32.8	3.18 (ddd, 6.5, 10.0, 17.5)	5‴	67.1	3.69 (dd, 10.5, 10.5)
		3.47 (ddd, 4.5, 10.0, 17.5)			4.37 (dd, 5.0, 10.5)
24	202.7	-			

**Table 10 molecules-25-00139-t010:** The ^1^H and ^13^C-NMR data for **10** in C_5_D_5_N

No.	δ_C_	δ_H_ (*J* in Hz)	No.	δ_C_	δ_H_ (*J* in Hz)
1	39.3	0.73 (m)	28	28.1	1.28 (s)
		1.54 (m, overlapped)			
2	26.8	1.83 (m, overlapped)	29	16.6	1.11 (s)
		2.20 (m)			
3	89.0	3.28 (dd, 4.0, 12.0)	30	17.4	0.97 (s)
4	39.7	-	1′	105.1	4.92 (d, 7.5)
5	56.4	0.66 (br. d, *ca.* 12)	2′	83.5	4.26 (dd, 7.5, 8.5)
6	18.4	1.37(m, overlapped)	3′	78.4	4.32 (m, overlapped)
		1.47 (m, overlapped)			
7	35.1	1.18 (m)	4′	71.7	4.35 (m, overlapped)
		1.46 (m, overlapped)			
8	40.0	-	5′	78.3	3.94 (m, H-5′)
9	50.2	1.36 (m, overlapped)	6′	62.9	4.35 (m, overlapped)
					4.57 (dd, 1.5, 11.5)
10	36.9	-	1″	106.1	5.39 (d, 7.5)
11	30.9	1.54 (m, overlapped)	2″	77.2	4.14 (dd, 7.5, 8.5)
		1.96 (m)			
12	70.1	4.19 (m)	3″	78.1	4.26 (m, overlapped)
13	49.5	2.03 (dd, 10.5, 10.5)	4″	71.7	4.14 (dd, 7.5, 8.5)
14	51.5	-	5″	78.2	3.93 (m)
15	30.7	0.98 (m)	6″	62.7	4.50 (m)
		1.54 (m, overlapped)			
16	26.7	1.37(m, overlapped)	1‴	98.0	5.10 (d, 8.0)
		1.83 (m, overlapped)			
17	52.1	2.58 (q like, *ca.* 11)	2‴	74.8	3.90 (dd, 8.0, 8.5)
18	16.0	0.92 (s)	3‴	79.5	4.17 (dd, 8.5, 8.5)
19	16.3	0.81 (s)	4‴	71.4	4.13 (m, overlapped)
20	83.2	-	5‴	76.6	4.05 (m)
21	21.7	1.58 (s)	6‴	70.2	4.27 (m, overlapped)
					4.77 (dd, 1.5, 11.5)
22	29.8	2.11 (m)	1⁗	106.1	4.91 (d, 7.5)
		2.75 (ddd, 4.0, 9.0, 16.5)			
23	32.8	3.18 (ddd, 5.5, 9.0, 16.5)	2⁗	74.9	4.02 (dd, 7.5, 8.5)
		3.45 (ddd, 4.0, 9.0, 16.5)			
24	202.6	-	3⁗	78.0	4.14 (m, overlapped)
25	144.4	-	4⁗	71.2	4.25 (dd, 8.0, 8.5)
26	125.5	5.51, 5.71 (both br. s)	5⁗	67.2	3.69 (dd, 10.5, 10.5)
					4.36 (m, overlapped)
27	17.9	1.83 (s)			

**Table 11 molecules-25-00139-t011:** The ^1^H and ^13^C-NMR data for **11** in C_5_D_5_N

No.	δ_C_	δ_H_ (*J* in Hz)	No.	δ_C_	δ_H_ (*J* in Hz)
1	38.5	0.89, 1.61 (both m)	23	23.2	2.34, 2.58 (both m)
2	23.8	1.80 (m)	24	125.9	5.32 (t, 6.5)
3	82.1	4.88 (dd, 4.0, 10.5)	25	131.1	
4	39.1		26	25.8	1.62 (s)
5	61.4	1.18 (d, 10.0)	27	17.9	1.66 (s)
6	67.3	4.32 (m, overlapped)	28	31.3	1.76 (s)
7	47.2	1.83 (m, overlapped)	29	17.0	1.35 (s)
		1.93 (dd, 11.0, 11.0)			
8	41.1		30	17.4	0.97 (s)
9	49.6	1.49 (m, overlapped)	1′	168.4	
10	39.0		2′	43.8	3.83 (s)
11	30.7	1.49 (m, overlapped)	3′	170.6	
		1.99 (m)			
12	70.1	4.19 (m)	1″	98.1	5.12 (d, 7.5)
13	49.1	1.97 (m, overlapped)	2″	74.9	3.94 (dd, 7.5, 7.5)
14	51.3		3″	79.2	4.18 (dd, 7.5, 9.0)
15	30.7	0.97 (m, overlapped)	4″	71.8	4.07 (dd, 9.0, 10.0)
		1.55 (m)			
16	26.6	1.31 (m)	5″	76.7	4.05 (m)
		1.81 (m, overlapped)			
17	51.6	2.55 (q like, *ca.* 10)	6″	69.3	4.26 (dd, 5.0, 10.5)
					4.70 (br. d, *ca.* 11)
18	17.5	1.05 (s)	1‴	104.7	5.00 (d, 5.5)
19	17.3	0.93 (s)	2‴	72.2	4.46 (dd, 5.5, 7.0)
20	83.4		3‴	74.1	4.23 (dd, 3.0, 7.0)
21	22.2	1.63 (s)	4‴	68.6	4.37 (m)
22	36.1	1.81 (m, overlapped)	5‴	65.7	3.79 (br. d, *ca.* 12)
		2.39 (m)			4.32 (dd, 3.0, 12.0)

**Table 12 molecules-25-00139-t012:** Inhibitory effects of **1**, **2**, **5**, **6**, and **8**–**10** on NO production in RAW 264.7 cells.

NO.	NRC (%)	NO.	NRC (%)
Normal	3.2 ± 0.6	5	36.8 ± 4.1 ***
Control	100 ± 4.2	6	51.3 ± 5.8 ***
DEX	75.3 ± 2.6 ***	8	44.6 ± 4.8 ***
1	62.4 ± 8.0 ***	9	30.1 ± 3.0 ***
2	33.1 ± 1.2 ***	10	39.4 ± 1.9 ***

Positive control: Dexamethasone (Dex). Nitrite relative concentration (NRC): percentage of control group (set as 100%). Values represent the mean ± SD of three determinations. *** *p* < 0.001 (Differences between compound-treated group and control group), *N* = 4. Final concentration was 50 μM for compounds **1**, **2**, **5**, **6**, **8**–**10**, and 1.0 μg/mL for positive control (Dex), respectively.

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
