# Peer review of "New Dammarane-Type Triterpenoid Saponins from Panax notoginseng Leaves and Their Nitric Oxide Inhibitory Activities"

_molecules, 2019, doi:10.3390/molecules25010139_

Round 1
Reviewer 1 Report
The manuscript describes 11 new dammarane-type triterpenoid saponins (1–11) from the leaves of Panax notoginseng and their inhibitory effects on NO production. Overall, the manuscript is well written and the new compounds are interested. The technology used in the structural elucidation are satisfactory.
Followings are minor issues that need to be addressed before acceptance in the Molecules.
Minor comments
Page 4: The stereochemistry of 1 (3β,12β,20S-) should be discussed in the text Page 4, lines 109-111: The sentence should be corrected. The main difference between 2 and 1: a-L-arabinofuranosyl group in 1 was replaced for β-D-xylopyranosyl group in 2.Author Response
Reviewer1
The manuscript describes 11 new dammarane-type triterpenoid saponins (1–11) from the leaves of Panax notoginseng and their inhibitory effects on NO production. Overall, the manuscript is well written and the new compounds are interested. The technology used in the structural elucidation are satisfactory.
Followings are minor issues that need to be addressed before acceptance in the Molecules.
Minor comments
Q1: Page 4: The stereochemistry of 1 (3β,12β,20S-) should be discussed in the text Page 4,
A1: Thank you for your suggestion, we revised it.
Q2: lines 109-111: The sentence should be corrected. The main difference between 2 and 1: a-L-arabinofuranosyl group in 1 was replaced for β-D-xylopyranosyl group in 2.
A2: Revised.
Reviewer 2 Report
New bioactive triterpenoid saponins from Panax notoginseng leaves are introduced. The complete and detailed report of isolated new compounds is provided. The experimental looks sound and as well the analysis/discussion, however many typos are observed and suggestions are given before acceptation for publication in Molecules:
Line 25-30 revise the end of abstract (suggestion)
“As results, 11 new dammarane-type triterpenoid saponins, notoginsenosides NL-A1-NL-A4 (1-4), NL-B1-NL-B3 (5-7), NL-C1-NL-C3 (8-10), together with NL-D (11) were isolated and their structures identified by using various spectrometric techniques and chemical reactions. In addition, compounds 1, 2, 5, 6, 8-10 were found to play important role in suppressing NO levels showing no cytotoxicity at 50 μM. All inhibitory activities were found to follow a dose-dependent manner.”
Line 40-43 Introductions (suggestion)
“For example, P. notoginseng was found to be the main ingredient in Xuesaitong injections and Xuesaitong capsules [2], Yun-Nan-Bai-Yao [3], as well as Xueshuantong injections and Xueshuantong capsules. As we know, triterpenoid saponins have made great contributions to the above-mentioned biological activities [2,3].”
Line 44-45 suggestion
“Since the ability of P. notoginseng to adapt to the environment has gradually declined through centuries of cultivation, the problem of continuous cropping has becoming more and more prominent, which resulted in the decrease of P. notoginseng root yields [4].”
Line 47 … long growth periods.
Line 52 “As we know, … “
Line 74, 186 “ and ether functions …”
Line 85 “HSQC-TOCSY experiment was performed. “
Line 92 “ three glycosyls were detailed assigned.
Line 129 “Forty four signals were …”
Line 169 “ spectral characteristics, …”
Line 176 “ in addition, the cross peaks …”
Line 180 “ for the aglycones of …”
Line 198 “ the following long …”
Line 209 “ disappeared in 9”
Line 237-238 suggestion
“Since there were more differences in 1-4 positions of two compounds, they might be affected by the configuration of C-3 or a malonyl substitution.”
Line 239 “correlations”
Line 240 “displayed”
Line 259 “ control group”
Line 253 “for compounds 1, 2, 5, 6, 8–10, and 1.0 μg/mL for positive …”
Line 295 “aliquot”
Line 308, 314, “HAc”
Line 531 please provide a brief description of hydrolysis because the reference 14 does not do it
Line 551 “draw”
Revise Conclusions writing, particularly second sentence of second paragraph
Author Response
New bioactive triterpenoid saponins from Panax notoginseng leaves are introduced. The complete and detailed report of isolated new compounds is provided. The experimental looks sound and as well the analysis/discussion, however many typos are observed and suggestions are given before acceptation for publication in Molecules:
Thank you for your patient and careful reading and modifying for the text. We revise the mistakes according to your well suggestions.
Q1: Line 25-30 revise the end of abstract (suggestion)
“As results, 11 new dammarane-type triterpenoid saponins, notoginsenosides NL-A1-NL-A4 (1-4), NL-B1-NL-B3 (5-7), NL-C1-NL-C3 (8-10), together with NL-D (11) were isolated and their structures identified by using various spectrometric techniques and chemical reactions. In addition, compounds 1, 2, 5, 6, 8-10 were found to play important role in suppressing NO levels showing no cytotoxicity at 50 μM. All inhibitory activities were found to follow a dose-dependent manner.”
A1: Revised.
Q2: Line 40-43 Introductions (suggestion)
“For example, P. notoginseng was found to be the main ingredient in Xuesaitong injections and Xuesaitong capsules [2], Yun-Nan-Bai-Yao [3], as well as Xueshuantong injections and Xueshuantong capsules. As we know, triterpenoid saponins have made great contributions to the above-mentioned biological activities [2,3].”
A2: Revised.
Q3: Line 44-45 suggestion
“Since the ability of P. notoginseng to adapt to the environment has gradually declined through centuries of cultivation, the problem of continuous cropping has becoming more and more prominent, which resulted in the decrease of P. notoginseng root yields [4].”
A3: Revised.
Q4: Line 47 … long growth periods.
A4: Revised.
Q5: Line 52 “As we know, … “
A5: Revised.
Q6: Line 74, 186 “ and ether functions …”
A6: Revised.
Q7: Line 85 “HSQC-TOCSY experiment was performed. “
A7: Revised.
Q8: Line 92 “ three glycosyls were detailed assigned.
A8: Revised.
Q9:Line 129 “Forty four signals were …”
A9: Revised.
Q10:Line 169 “ spectral characteristics, …”
A10: Revised.
Q11:Line 176 “ in addition, the cross peaks …”
A11: Revised.
Q12:Line 180 “ for the aglycones of …”
A12: Revised.
Q13:Line 198 “ the following long …”
A13: Revised.
Q14:Line 209 “ disappeared in 9”
A14: Revised.
Q15:Line 237-238 suggestion
“Since there were more differences in 1-4 positions of two compounds, they might be affected by the configuration of C-3 or a malonyl substitution.”
A15: Revised.
Q16:Line 239 “correlations”
A16: Revised.
Q17:Line 240 “displayed”
A17: Revised.
Q18:Line 259 “ control group”
A18: Revised.
Q19:Line 253 “for compounds 1, 2, 5, 6, 8–10, and 1.0 μg/mL for positive …”
A19: Revised.
Q20:Line 295 “aliquot”
A20: Revised.
Q2:Line 308, 314, “HAc”
A21: Revised.
Q22:Line 531 please provide a brief description of hydrolysis because the reference 14 does not do it
A22: Revised.
Q23: Line 551 “draw”
A23: Revised.
Q24: Revise Conclusions writing, particularly second sentence of second paragraph
A24: Thank you for your suggestion, we rewrite the conclusions.
Reviewer 3 Report
The Authors described the investigations on isolation and identification new dammarane-type triterpenoid saponins from Panax notoginseng leaves and prediction of their nitric oxide inhibitory activities. The investigations on contents of biologically active compounds in plant extracts and their biological activity are an important aspect of scientific research. The advantage of the described investigations is isolation and identification by various spectral techniques such as IR, MS, H NMR and C NMR of the structure of eleven dammarane-type triterpenoid saponins from Panax notoginseng leaves . Authors additionally confirmed the important role of some of isolated compounds in suppressing NO levels without cytotoxicity in the concebtration of 50 µM. In my opinion the topic of the manuscript is interesting and the manuscript is worth publishing. The conducted research may contribute in a certain degree to progress in the search for new effective anti-inflammatory drugs.
In order to complete the presentation of the obtained results, I would suggest including an example chromatogram obtained for the separation of investigated compounds.
The Authors also should clearly highlight the novelty, advantages and practical usefulness of their research especially comparison to previously published results e.g. ref. 6 (Li, J.; Wang, R.F.; Zhou, Y.; Hu, H.J.; Yang, Y.B.; Yang, L.; Wang, Z.T. Dammarane-type triterpene oligoglycosides from the leaves and stems of Panax notoginseng and their antiinflammatory activities. J. Ginseng Res. 589 2019, 43, 377–384.)
Reviewer 4 Report
The paper describes the isolation and spectroscopic identification of new dammarane-type triterpenoid saponins from panax notoginseng leaves and their nitric oxide inhibitory activities. The paper could be published subject to revisions as indicated below:
Major improvement of the English text should be made. Page 2, lines 47-48“Researches showed that P. notoginseng leaves were rich in dammarane-type triterpenoid saponins (PNS) [5-7].”
Only three references from 2019 were selected. The authors should provide a more comprehensive literature search.
Figure 1.Compounds 1, 2, 3 and 4 are hydroperoxides. The authors should provide NMR spectral identification of the O-O-H proton. There is recent literature of how to use NMR methods for the identification of hydroperoxide protons and carbons.
Figure 2.The authors should explain why C6 and the attached protons do not indicate key proton 1H-1H COSY and 1H-13C HMBC connectivities.
Table 1.The 1H NMR data should also be included.
Page 7, lines 138-140.“The 3-OH group was established as β-oriented (equatorial) from the NOESY correlations beween δH 4.01 (H-12) and δH 1.38 (H-9); δH 0.78 (H-5) and δH 1.38 (H-9), 4.81 (H-3) and vicinal coupling constants of H-3 with H2-2 (3J = 4.5, 11.5 Hz).”
Reference should be given to the original literature about using the above methods for establishing the equatorial orientation of the 3-OH group.
The authors used C5D5N as solvent for the NMR experiments. Due to the basicity of the solvent, the OH resonances of the sugars appear as a broad signal. It is not clear why the authors did not use DMSO-d6 which, according to recent literature, are very effective in reducing 1H exchange rate and, thus, sharp OH resonances can be observed. No details are provided of the experimental NMR parameters used such as number of scans, acquisition time, number of increments, total experimental time etc. The authors provided 74 figures in the Supplementary Information the majority of which are NMR spectra. Those spectra, however, are not useful since there are no assignments at least of the major resonances.Author Response
The paper describes the isolation and spectroscopic identification of new dammarane-type triterpenoid saponins from panax notoginseng leaves and their nitric oxide inhibitory activities. The paper could be published subject to revisions as indicated below:
Major improvement of the English text should be made. Page 2, lines 47-48
Q1: “Researches showed that P. notoginseng leaves were rich in dammarane-type triterpenoid saponins (PNS) [5-7].” Only three references from 2019 were selected. The authors should provide a more comprehensive literature search.
A1: Thank you for your well suggestions. We add other related references in the introduction part.
Q2: Figure 1. Compounds 1, 2, 3 and 4 are hydroperoxides. The authors should provide NMR spectral identification of the O-O-H proton. There is recent literature of how to use NMR methods for the identification of hydroperoxide protons and carbons.
A2: According to your advice, we revised the description of compound 1.
Q3: Figure 2. The authors should explain why C6 and the attached protons do not indicate key proton 1H-1H COSY and 1H-13C HMBC connectivities.
A3: In the first, we had drawn the 1H-1H COSY of Second, we only draw the main long range correlations from H to C for all compounds in Figure 2. According to the long range correlations from H3-19, 28, 29 to C-5; H3-18, to C-7 as well as 1H-1H COSY between H2-6 with H-5 and H2-7, we can connect A, B, and C rings together. Though the long range correlation from H2-6 to C-3, C-8, and C-10 were observed in its HMBC spectrum, it is not necessary to dram them in Figure 2.
Q4: Table 1. The 1H NMR data should also be included.
A4: The 1H NMR data of all compounds were provided in the “Extraction and Isolation” part. If we report 1H and 13C NMR data of all 11 new compounds in Table, there will be 11 tables in the text at least, which will make the type difficult.
Q5: Page 7, lines 138-140. “The 3-OH group was established as β-oriented (equatorial) from the NOESY correlations beween δH 4.01 (H-12) and δH 1.38 (H-9); δH 0.78 (H-5) and δH 1.38 (H-9), 4.81 (H-3) and vicinal coupling constants of H-3 with H2-2 (3J = 4.5, 11.5 Hz).” Reference should be given to the original literature about using the above methods for establishing the equatorial orientation of the 3-OH group.
A5: We determined the configuration of 3-OH as following: (1) Since the NMR data for aglycone of 4 were remarkably reminiscent of those of compounds 1, 2, and 3 besides C-2-4, we could draw the conclusion that the H-12 was α-oriented; (2) Since the correlations beween dH 4.01 (H-12) and dH 1.38 (H-9); dH 0.78 (H-5) and dH 1.38 (H-9), 4.81 (H-3) were observed in the NOESY, we could draw the conclusion that the H-3 was α-oriented, too. Then 3-OH group was established as β-oriented; (3) Since vicinal coupling constants of H-3 with H2-2 was 4.5 and 11.5 Hz, we can draw the conclusion that H-3 was in axial bond (As we know, if the H-3 was in equatorial bond, the vicinal coupling constants of H-3 with H2-2 was about 3 and 5 Hz ). Then the 3-OH was in equatorial bond. It is not necessary to cite the literature to prove it.
Q6: The authors used as solvent for the NMR experiments. Due to the basicity of the solvent, the OH resonances of the sugars appear as a broad signal. It is not clear why the authors did not use DMSO-d6 which, according to recent literature, are very effective in reducing 1H exchange rate and, thus, sharp OH resonances can be observed.
A6: Since the solubility of isolated dammarane-type triterpenoid saponins is not well in DMSO-d6, but very well in C5D5N, we selected C5D5N as solvent for the NMR experiments. On the other hand, some proton signals of dammarane-type triterpenoid saponins display at d 2.0-3.0, which will be overlapped by the residue proton signal of DMSO-d6. Consequently, we can not use DMSO-d6 as solvent to determine NMR data of dammarane-type triterpenoid saponins.
Q7: No details are provided of the experimental NMR parameters used such as number of scans, acquisition time, number of increments, total experimental time etc.
A7: Since the NMR parameters used such as number of scans, acquisition time, number of increments, total experimental time etc. for every compound is difference, we can not provide unified NMR parameters for all compounds. But the parameters of every NMR experiment was provided in the 74 figures in the supplementary information.
A8: The authors provided 74 figures in the Supplementary Information the majority of which are NMR spectra. Those spectra, however, are not useful since there are no assignments at least of the major resonances.
A8: The purpose of attaching 74 figures in the supplementary information is to provide evidence for the reported data and make it easy for readers to read. Since we reported the 1H and 13C NMR data of all compounds in the text, .Though there are no assignments of resonances, there is no necessary for further assigning the major resonances in the accompanying drawings.
Round 2
Reviewer 4 Report
Q2: Figure 1. Compounds 1, 2, 3 and 4 are hydroperoxides. The authors should provide NMR spectral identification of the O-O-H proton. There is recent literature of how to use NMR methods for the identification of hydroperoxide protons and carbons.
The main argument of the authors is the deshielding of the C-25. However, there are recent NMR methods for the unequivocal identification of C-O-O-H protons through 1H-13C HMBC and 1D NOESY experiment. Reference also should be given to recent NMR literature. Further, on page 4, line 99” “The chemical shift of C-25 shifted to lower field about 11 comparing…”.
The chemical shift is not shifted, it is the resonance that is shifted; the term low field should be avoided; about 11 there is something missing here (ppm?).
A4: The 1H NMR data of all compounds were provided in the “Extraction and Isolation” part. If we report 1H and 13C NMR data of all 11 new compounds in Table, there will be 11 tables in the text at least, which will make the type difficult.
Inclusion of 1H data in the two 13C tables will not increase the number of Tables. Furthermore, common 1H and 13C data will facilitate the incorporation of Tables into NMR data banks.
A5: We determined the configuration of 3-OH as following: (1) Since the NMR data for aglycone of 4 were remarkably reminiscent of those of compounds 1, 2, and 3 besides C-2-4, we could draw the conclusion that the H-12 was α-oriented; (2) Since the correlations beween dH 4.01 (H-12) and dH 1.38 (H-9); dH 0.78 (H-5) and dH 1.38 (H-9), 4.81 (H-3) were observed in the NOESY, we could draw the conclusion that the H-3 was α-oriented, too. Then 3-OH group was established as β-oriented; (3) Since vicinal coupling constants of H-3 with H2-2 was 4.5 and 11.5 Hz, we can draw the conclusion that H-3 was in axial bond (As we know, if the H-3 was in equatorial bond, the vicinal coupling constants of H-3 with H2-2 was about 3 and 5 Hz ). Then the 3-OH was in equatorial bond. It is not necessary to cite the literature to prove it.
The arguments of the authors are correct. However, the suggested methodology (use of vicinal coupling constants of H-3 and H-2 to determine the configuration of 3-OH) is based on the existing literature. Reference should be given.
In conclusion, the authors should further revise their article before acceptance.
